# The Elusive Pursuit of Reproducing PATE-GAN: Benchmarking, Auditing, Debugging

**Georgi Ganev**                                                   *georgi.ganev.16@ucl.ac.uk*
*University College London and SAS*

**Meenatchi Sundaram Muthu Selva Annamalai**                       *meenatchi.annamalai.22@ucl.ac.uk*
*University College London*

**Emiliano De Cristofaro**                                         *emilianodc@cs.ucr.edu*
*University of California, Riverside*

**Reviewed on OpenReview:** *https://openreview.net/forum?id=wcxrJcJ7vq*

## Abstract

Synthetic data created by differentially private (DP) generative models is increasingly used in real-world settings. In this context, PATE-GAN has emerged as one of the most popular algorithms, combining Generative Adversarial Networks (GANs) with the private training approach of PATE (Private Aggregation of Teacher Ensembles).

In this paper, we set out to reproduce the utility evaluation from the original PATE-GAN paper, compare available implementations, and conduct a privacy audit. More precisely, we analyze and benchmark six open-source PATE-GAN implementations, including three by (a subset of) the original authors. First, we shed light on architecture deviations and empirically demonstrate that none reproduce the utility performance reported in the original paper. We then present an in-depth privacy evaluation, which includes DP auditing, and show that *all implementations leak more privacy than intended*. Furthermore, we uncover *19 privacy violations* and 5 other bugs in these six open-source implementations. Lastly, our codebase is available from: https://github.com/spalabucr/pategan-audit.

## 1 Introduction

Privacy-preserving synthetic data has been increasingly adopted to share data within and across organizations while reducing privacy risks. The intuition is to train a generative model on the real data, draw samples from the model, and create new (synthetic) data points. As the original data may contain sensitive and/or personal information, synthetic data can be vulnerable to membership/property inference, reconstruction attacks, etc. (Hayes et al., 2019; Hilprecht et al., 2019; Chen et al., 2020; Stadler et al., 2022; Annamalai et al., 2024a; Ganev & De Cristofaro, 2023). Thus, models should be trained to satisfy robust definitions like Differential Privacy (DP) (Dwork et al., 2006; Dwork & Roth, 2014), which bounds the privacy leakage from the synthetic data. Combining generative models with DP has been advocated for or deployed by government agencies (NIST, 2018; Abowd et al., 2022; ONS, 2023; Hod & Canetti, 2024), regulatory bodies (ICO, 2023; FCA, 2024), and non-profit organizations (UN, 2023; OECD, 2023).

Numerous DP generative models have been proposed (Zhang et al., 2017; Xie et al., 2018; Zhang et al., 2018; Jordon et al., 2019; McKenna et al., 2021; Zhang et al., 2021; McKenna et al., 2022), etc. Alas, new DP models are often published without a public/thoroughly reviewed codebase. Correctly implementing DP mechanisms and effectively communicating their properties are often very challenging tasks in the real world (Cummings et al., 2021; Houssiau et al., 2022b). This prompts the need for rigorous reproducibility efforts and auditing studies to confirm the utility and privacy claims of state-of-the-art DP generative models.

|  | original | updated | synthcity | turing | borealis | smartnoise |
|---|---|---|---|---|---|---|
| **AUROC** $\Delta$ | -24.57% | -50.80% | -33.76% | -77.38% | -27.50% | -26.92% |

Table 1: Average reduction in AUROC (the random score 0.5 used as baseline) from the original paper (Jordon et al., 2019) across twelve classifiers and four datasets ($\epsilon = 1$) for the six implementations.

| | | | **Privacy Violations** | | | | | **Other Bugs** | |
|---|---|---|---|---|---|---|---|---|---|
| **Implement.** | PATE | Data Partition | Moments Accountant | Metadata | Laplace Noise | $\delta$ Scale | Labels Distribution | Teachers | Processing |
| original | ✓ | ✓ | | ✓ | ✓ | ✓ | ✓ | | |
| updated | | ✓ | | ✓ | ✓ | | | ✓ | ✓ |
| synthcity | | ✓ | ✓ | ✓ | | | | ✓ | |
| turing | ✓ | ✓ | | | ✓ | ✓ | | | ✓ |
| borealis | | | ✓ | ✓ | | | | | ✓ |
| smartnoise | | | ✓ | | | | | | |

Table 2: Overview of the privacy violations and bugs found in each implementation.

In this paper, we focus on PATE-GAN (Jordon et al., 2019), which combines Private Aggregation of Teacher Ensembles (PATE) (Papernot et al., 2017; 2018) and Generative Adversarial Networks (GANs) (Goodfellow et al., 2014) to train a generator in a privacy-preserving way using $k$ teachers-discriminators and a student-discriminator. PATE-GAN, published at ICLR'19, is among the two most popular DP generative models for tabular data and the most popular deep learning model (with over 800 citations as of February 2025), remaining widely used, extensively studied, and re-implemented by researchers and practitioners. Moreover, PATE-GAN is very competitive vs. state-of-the-art models in terms of utility, privacy, and fairness, particularly for high dimensional datasets (Rosenblatt et al., 2020; Ganev et al., 2022; 2024; Du & Li, 2024). We examine six public PATE-GAN implementations, including three by the original authors:

1. original[1], the first public code release in 2019 alongside the paper (Jordon et al., 2019);
2. updated[2], released in 2021 and linked to from the latest version of the paper (Jordon et al., 2019);
3. synthcity[3], the most recent implementation, included in the popular synthetic data benchmarking library synthcity (Qian et al., 2023);
4. turing[4], developed by the Alan Turing Institute, part of the TAPAS library (Houssiau et al., 2022a);
5. borealis[5], part of Borealis AI's toolbox for private generation of synthetic data;
6. smartnoise[6], included in OpenDP's popular library for DP synthetic data (OpenDP, 2021).

Our reproducibility study has two main objectives. First, we set to reproduce the utility performance reported in the original paper (Jordon et al., 2019) by studying and benchmarking the six implementations. Second, we empirically estimate PATE-GAN's privacy guarantees using DP auditing tools. In our experiments, we use all four publicly available tabular datasets (two Kaggle and two UCI) used in the original evaluation, a common image dataset (MNIST), and create a worst-case dataset as part of the DP auditing tests.

Our experimental evaluation yields several main findings:

- We fail to reproduce the utility performance reported in the original paper (Jordon et al., 2019) in any of the six implementations we benchmark, with an average utility drop ranging from 25% to 77% across the four datasets (see Table 1);
- All six implementations leak more privacy than they should – i.e., the empirical privacy estimates we obtain using black-box membership inference attacks are worse than the theoretical differential privacy bounds;

---

[1] https://bitbucket.org/mvdschaar/mlforhealthlabpub/src/master/alg/pategan/PATE_GAN.py
[2] https://github.com/vanderschaarlab/mlforhealthlabpub/blob/main/alg/pategan/pate_gan.py
[3] https://github.com/vanderschaarlab/synthcity/blob/main/src/synthcity/plugins/privacy/plugin_pategan.py
[4] https://github.com/alan-turing-institute/reprosyn/blob/main/src/reprosyn/methods/gans/pate_gan.py
[5] https://github.com/BorealisAI/private-data-generation/blob/master/models/pate_gan.py
[6] https://github.com/opendp/smartnoise-sdk/blob/main/synth/snsynth/pytorch/nn/pategan.py

- As summarized in Table 2, we identify 19 privacy violations and 5 other bugs, predominantly in how PATE is implemented, e.g., partitioning and feeding data to the teachers-discriminators, tracking the privacy budget during model training, etc.

To facilitate open research and robust privacy (re-)implementations, we release our codebase, including the utility benchmark and privacy auditing tools; see https://github.com/spalabucr/pategan-audit.

## 2 Preliminaries

**Notation.** In the rest of the paper, we denote a dataset containing $N$ $d$-dimensional samples as $\mathcal{D} = \{(\boldsymbol{x_i}, y_i)\}_{i=1}^N$, where $\boldsymbol{x_i} \in \mathcal{X}$ (the feature space) and $y_i \in \mathcal{Y}$ (the label space).

**Differential Privacy (DP).** DP is a mathematical formalization of privacy that limits the influence of the input data points on the output of a function. In the context of machine learning, DP bounds the probability of distinguishing whether any particular record was used to train a model. Formally, a randomized algorithm $\mathcal{A}$ satisfies $(\epsilon, \delta)$-DP if, for all $S \subseteq \text{Range}(\mathcal{A})$ and for all neighboring datasets, $\mathcal{D}$ and $\mathcal{D}'$ differing in a single data record, it holds that (Dwork et al., 2006; Dwork & Roth, 2014):

$$\Pr[\mathcal{A}(\mathcal{D}) \in S] \le e^\epsilon \cdot \Pr[\mathcal{A}(\mathcal{D}') \in S] + \delta$$

The privacy budget $\epsilon$ is a positive number quantifying the privacy leakage (the lower, the better), while $\delta$ is a very small number representing the probability of *failure*. The *post-processing* property guarantees that a DP mechanism's output can be used arbitrarily without additional privacy leakage.

**PATE.** The Private Aggregation of Teacher Ensembles (PATE) framework (Papernot et al., 2017; 2018) trains a DP discriminative model using semi-supervised learning techniques. A model trainer is assumed to have access to a *private* dataset and an unlabelled *public* dataset. First, the private dataset is separated into $k$ *disjoint* partitions, and a teacher-classifier is trained on each partition (without privacy), then, the teachers predict the labels of the public dataset. Next, the predictions are noisily aggregated through the Laplace mechanism (Dwork et al., 2006) to get DP labels. Finally, a student-classifier is trained on the pairs of public records and noisy (DP) labels. The student-classifier can be released as it satisfies DP through the post-processing property (since it was trained on public data and DP labels).

**PATE-GAN.** In the context of synthetic data, a generative model $G$ is typically fitted on $\mathcal{D}$ to capture a probability representation and is later sampled to generate new data (of size $N'$), $\mathcal{S} \sim G(N')$. PATE-GAN (Jordon et al., 2019) incorporates PATE into the standard mini-max generator vs. discriminator training in a Generative Adversarial Network (GAN) (Goodfellow et al., 2014). We report its pseudo-code in Algorithm 1 in Appendix A.1. As before, $\mathcal{D}$ is first separated into $k$ disjoint partitions. In each iteration, $k$ teachers-discriminators, $T_1 \ldots T_k$, are trained on their corresponding data partition. Instead of using a public dataset, in PATE-GAN, the generator $G$ generates samples that are labeled by the teachers as "real" or "fake" through the PATE mechanism. The student-discriminator $S$ is then trained on these generated samples and noisy (DP) labels. Finally, the generator is trained by minimizing the loss on the student. Since the generator is only exposed to the student-discriminator, which in turn sees only "fake" generated samples and noisy (DP) labels, PATE-GAN satisfies DP through the post-processing property. Throughout the training, the privacy budget spent in each iteration is tracked by an adjusted moments accountant (Abadi et al., 2016). In the accountant's calculations (see lines 16-19 in Algorithm 1), $\lambda$ denotes the scale of the Laplace mechanism added by PATE when aggregating the teachers' "fake" and "real" votes ($n_0$ and $n_1$, respectively), while $L$ is the number of moments. As explained in the original paper (Jordon et al., 2019), finding the right balance between the noise level controlled by $\lambda$ (where larger values result in less added noise) and the number of teachers $k$ is key to achieving meaningful aggregation. Finally, in Appendix A.2, we discuss discrepancies between PATE-GAN and PATE (Papernot et al., 2017).

**DP Auditing.** This entails *empirically* estimating the privacy leakage from a DP model, denoted as $\epsilon_{emp}$, and comparing it with the *theoretical* privacy budget, $\epsilon$. One key goal is for the estimates to be tight (i.e., $\epsilon_{emp} \approx \epsilon$), so that the audit can be effectively used to validate the DP implementation or detect DP violations in case $\epsilon_{emp} > \epsilon$ (Bichsel et al., 2018; 2021; Niu et al., 2022). To estimate $\epsilon_{emp}$, we use membership inference attacks (MIAs) (Shokri et al., 2017; Hayes et al., 2019), whereby an adversary attempts to infer

whether a target record was part of the training data (i.e., $(\boldsymbol{x}_T, y_T) \in \mathcal{D}$), which closely matches the DP definition. The MIA process is run repeatedly as a distinguishing game; we randomly select between $\mathcal{D}$ and $\mathcal{D}' = \mathcal{D} \setminus (\boldsymbol{x}_T, y_T)$, pass it to the adversary, and get their predictions $Pred = \{pred_1, pred_2, ...\}$ and $Pred' = \{pred'_1, pred'_2, ...\}$. These yield a false positive rate $\alpha$ and a false negative rate $\beta$. 95% confidence upper bounds for the $\alpha$ and $\beta$ ($\overline{\alpha}$ and $\overline{\beta}$) are typically calculated using Clopper-Pearson intervals (Clopper & Pearson, 1934), but recent work has shown that using Bayesian credible intervals instead can dramatically improve the $\epsilon_{emp}$ estimates (Zanella-Béguelin et al., 2023). However, Nasr et al. (2023) note that using Bayesian credible intervals may not produce statistically valid lower bounds, and therefore we mainly use Clopper-Pearson bounds throughout the paper. Finally, the upper bounds are converted into the lower bound $\epsilon_{emp}$, as done in (Nasr et al., 2021):

$$\epsilon_{emp} = \max \left\{ \log \left( (1 - \overline{\alpha} - \delta) / \overline{\beta} \right), \log \left( (1 - \overline{\beta} - \delta) / \overline{\alpha} \right), 0 \right\}$$

## 3 Related Work

**DP Generative Models Benchmarks.** There are multiple DP generative models for synthetic tabular data, including copulas (Li et al., 2014; Gambs et al., 2021), graphical models (Zhang et al., 2017; McKenna et al., 2021; Cai et al., 2021; Mahiou et al., 2022), workload/query-based (Vietri et al., 2020; Aydore et al., 2021; Liu et al., 2021; Vietri et al., 2022; McKenna et al., 2022; Maddock et al., 2023a), and deep generative models like VAEs (Acs et al., 2018; Abay et al., 2018), GANs (Xie et al., 2018; Zhang et al., 2018; Jordon et al., 2019; Alzantot & Srivastava, 2019; Frigerio et al., 2019; Long et al., 2021), and others (Zhang et al., 2021; Ge et al., 2021; Truda, 2023; Vero et al., 2024).

A few benchmarking studies (Rosenblatt et al., 2020; Tao et al., 2022; Ganev et al., 2024; Du & Li, 2024) compare PATE-GAN to other DP generative models. Although PATE-GAN is not the best-performing model at fidelity and in similarity-based evaluations, it is among the best at downstream classification (for `original`, `synthcity`, `smartnoise`) compared to PrivBayes (Zhang et al., 2017), PrivSyn (Zhang et al., 2021), MST (McKenna et al., 2021), DPGAN (Xie et al., 2018), DP-WGAN (Alzantot & Srivastava, 2019), and TableDiffusion (Truda, 2023), especially on datasets with large number of columns (Rosenblatt et al., 2020; Ganev et al., 2024; Du & Li, 2024). On the other hand, Tao et al. (2022) claim that PATE-GAN (`smartnoise`) cannot beat simple baselines, although they only use narrow datasets (fewer than 25 columns).

None of these benchmarks use the datasets from the original paper (Jordon et al., 2019). Moreover, when developing new PATE-GAN implementations, the authors do not compare them to previous ones. Therefore, this prompts the need to evaluate them against each other and check how closely they resemble the original model (Jordon et al., 2019) in terms of architecture and utility performance.

**DP Generative Models Auditing.** Recent research on DP auditing has focused on discriminative models trained with DP-SGD (Abadi et al., 2016) in both central (Jayaraman & Evans, 2019; Jagielski et al., 2020; Tramer et al., 2022; Nasr et al., 2021; 2023; Zanella-Béguelin et al., 2023; Steinke et al., 2023; Kong et al., 2024; Annamalai & De Cristofaro, 2024; Cebere et al., 2024) and federated (Maddock et al., 2023b; Andrew et al., 2024) settings. For generative models, Houssiau et al. (2022a) loosely audit MST (McKenna et al., 2021) using Querybased, a black-box attack which runs a collection of random queries, while Annamalai et al. (2024b) are the first to tightly audit PrivBayes (Zhang et al., 2017), MST (McKenna et al., 2021), and DP-WGAN (Alzantot & Srivastava, 2019) (and detect several DP-related bugs) by proposing implementation specific white-box attacks and worst-case data records. Also, Lokna et al. (2023) find floating points vulnerabilities in MST (McKenna et al., 2021) and AIM (McKenna et al., 2022).

PATE-GAN has been mostly overlooked by previous work on DP auditing. Using a novel shadow model-based black-box attack and manually crafted worst-case target, GroundHog, Stadler et al. (2022) show that PATE-GAN (`original`) leaks more privacy than it should, while van Breugel et al. (2023) show that PATE-GAN (`synthcity`) is less private than non-DP models like CTGAN (Xu et al., 2019) and ADS-GAN (Yoon et al., 2020) using a density-based MIA. However, neither work explores the underlying reasons. In this work, we examine six PATE-GAN implementations to determine whether the privacy leakage is unique to a single implementation and dig deeper into the underlying reasons behind DP bugs/violations. This number is comparable to, and often exceeds, the number of implementations studied in previous privacy attacks vs.

| | (Jordon et al., 2019) | original | updated | synthcity | turing | borealis | smartnoise |
|---|---|---|---|---|---|---|---|
| Using PATE? | ✓ | ✗ | ✓ | ✓ | ✗ | ✓ | ✓ |
| Teachers | $N/1{,}000$ | - | 10 | 10 | - | 10 | $N/1{,}000$ |
| $\lambda$ | - | - | 1 | 1e-3 | - | 1e-4 | 1e-3 |
| $\alpha$ size ($L$) | - | - | 20 | 100 | - | 100 | 100 |
| $\delta$ | 1e-5 | 1e-5 | 1e-5 | $1/(N\sqrt{N})$ | 1e-5 | 1e-5 | $1/(N\sqrt{N})$ |
| Framework | - | TensorFlow | TensorFlow | PyTorch | TensorFlow | PyTorch | PyTorch |
| Optimizer | Adam | Adam | RMSProp | Adam | Adam | Adam | Adam |
| Learning Rate | 1e-4 | 1e-4 | 1e-4 | 1e-4 | 1e-4 | 1e-4 | 1e-4 |
| Batch Size | 64 | 128 | 64 | 200 | 128 | 64 | 64 |
| Max Iterations | -[†] | 10,000 | -[†] | 1,000 or -[†] | 100 | -[†] | -[†] |
| Teachers Iterations | 5 | -[‡] | 1 | 1[§] | -[‡] | 5 | 5 |
| Student Iterations | 5 | 1 | 5 | 10[§] | 1 | 5 | 5 |
| Generator Iterations | 1 | 1 | 1 | 10[§] | 1 | 1 | 1 |
| Teachers Layers | 1 layer | -[‡] | $\{1\}$ | $\{1\}$ | -[‡] | $\{d/2, 1\}$ | $\{2d/3, d/3, 1\}$ |
| Student Layers | 3 layers | $\{d, d, 1\}$ | $\{d, 1\}$ | $\{100, 1\}$ | $\{d, d, 1\}$ | $\{d/2, 1\}$ | $\{2d/3, d/3, 1\}$ |
| Noise Dimension | - | $d/4$ | $d$ | $d$ | $d/4$ | $d/4$ | 64 |
| Generator Layers | $\{d, d/2, d\}$ | $\{d, d, d\}$ | $\{4d, 4d, d\}$ | $\{100, d\}$ | $\{d, d, d\}$ | $\{2d, d\}$ | $\{64, 64, d\}$ |

[†]Until there is no available privacy budget. [‡]There are no teacher-/student-discriminators (only a single discriminator). [§]Epochs.

Table 3: Architectures and hyperparameters of the six PATE-GAN implementations ($N$ and $d$ are the number of data records and columns; the values in {} denote the width of the corresponding layer).

DP generative models for tabular data – six DP implementations (Annamalai et al., 2024b), three (Stadler et al., 2022), and two (Houssiau et al., 2022a; van Breugel et al., 2023; Annamalai et al., 2024a).

**Reproducibility Studies.** Overall, papers aimed at reproducing the computational experiments of previously published work not only verify their empirical results but also ensure the reliability and trustworthiness of their claims. This is particularly important in fields like machine learning and security, where individual privacy may be at risk. For instance, recent work in this space includes (Nokabadi et al., 2024; Garg & Tiwari, 2024; Feng et al., 2024; Chaudhuri et al., 2024).

## 4 PATE-GAN Implementations

This section reviews the six implementations under study, as also summarized in Table 3. Algorithm 1 in Appendix A.1 reports the PATE-GAN algorithm as presented in the paper (Jordon et al., 2019), while we highlight any deviations in the six implementations in Appendix E.

**Architectures.** In the first column of Table 3, we list the architecture and hyperparameters of the PATE-GAN algorithm as explicitly stated in the original paper (Jordon et al., 2019). However, we find that `original`, as well as `turing`, do not actually implement the PATE framework or a moments accountant; rather, there is a single discriminator (not $k$ teachers and a student; see lines 2, 9-10, and 17 in Algorithm 1) that observes all the data and is trained for a given number of iterations since the privacy budget spent is not tracked (lines 4, 18-21, and 29). This is a serious deviation that can compromise privacy. Also, `updated` and `synthcity`, released by a subset of the original authors, do not use neural networks as teachers, but use Logistic Regression classifiers that are (re-)fitted from scratch on every iteration (lines 8-10): while this does not violate privacy, it might negatively affect the model's utility. Moreover, `updated` uses the RMSProp optimizer instead of Adam. Finally, `borealis` and `smartnoise` resemble the original algorithm the closest, although they change the networks' depth, with `smartnoise` opting for teachers with three layers instead of the default one. Also, all implementations have different network depths and noise dimensions, mostly depending on the input data.

**Data Support and Processing.** The implementations also differ in the kind of data types they support and how they process the input data. For instance, `original` only runs on numerical and binary features, i.e., categorical columns can have at most two distinct categories and are treated as numerical data. Only

| Dataset | Paper (Jordon et al., 2019) | `original` | `updated` | `synthcity` | `turing` | `borealis` | `smartnoise` |
|---|---|---|---|---|---|---|---|
| Kaggle Credit | 0.8737 | 0.5000 | 0.5000 | 0.5172 | 0.5000 | 0.6300 | 0.7358 |
| Kaggle Cervical Cancer | 0.9108 | 0.9265 | 0.8739 | 0.8584 | 0.5000 | 0.9431 | 0.8025 |
| UCI ISOLET | 0.6399 | 0.8342 | 0.5861 | 0.6292 | 0.5000 | 0.6219 | 0.6152 |
| UCI Epileptic Seizure | 0.7681 | 0.6867 | 0.6187 | 0.7165 | 0.7425 | 0.6730 | 0.6963 |

Table 4: Average AUROC scores of the different implementations over the 12 classifiers ($\epsilon = 1$).

two implementations, `turing` and `synthcity`, preserve the original data types (such as integers) in the newly generated synthetic data.

In terms of data processing, `original` and `turing` scale the numerical data between 0 and 1 while `smartnoise` between -1 and 1. Also, `synthcity` transforms the numerical data and centers it around 0 by fitting a Bayesian Gaussian Mixture model on every column and using a standard scaler on the clusters. On the other hand, `updated` and `borealis` expect data that has already been processed/scaled and, consequently, do not return synthetic data in the scale of the input data (we adjust these models and use min-max scaling, similarly to the others). Apart from `smartnoise`, none of the implementations extract the data bounds in a DP way, which has been proven to leak privacy (Stadler et al., 2022; Annamalai et al., 2024b). (Note, in fact, that `turing` allows for the bounds to be passed as input).

***Remarks.*** The main goals of this paper are to reproduce the empirical experiments in the original paper (Jordon et al., 2019) and to test the privacy properties of the implementations as they were released, without modifying them. We do so as these implementations are in the public domain and have been adopted and used in practice (e.g., both `synthcity` and `smartnoise` libraries have average monthly downloads exceeding 4,500).[7] Specifically, we assume that the architectures, hyperparameters, and data processing decisions the authors have chosen for their respective implementations are optimal and do not change them unless stated otherwise. While we leave fixing discrepancies and bugs to future work, we have contacted the authors regarding these issues (see Sections 6.3 and 7) and have offered to assist them with addressing the bugs.

## 5   Utility Benchmark

In this paper, we reproduce the utility experiments on all public datasets from the original paper (Jordon et al., 2019), i.e., Kaggle Credit, Kaggle Cervical Cancer, UCI ISOLET, and UCI Epileptic Seizure (we also use MNIST; see Appendix B.1).

**Setup.** We use the evaluation criteria from (Jordon et al., 2019): for every synthetic dataset, we fit 12 classifiers (see Appendix B.2) and report the Area Under the Receiver Operating Characteristic curve (AUROC) and the Area Under the Precision-Recall Curve (AUPRC) scores. For the sake of comparison, as done by the original authors in `updated`, we consider the *best* of the scores from 25 synthetic datasets, training five models and generating five synthetic datasets per model. We use an 80/20 split, i.e., using 80% of the records in the datasets to train the predictive/generative models and 20% for testing. Finally, we use two training-testing settings:

- *Setting A*: train on the real training dataset, test on the real testing dataset;
- *Setting B*: train on the synthetic dataset, test on the real testing dataset (as done in (Esteban et al., 2017));

**Hyperparameters.** We set $\delta = 10^{-5}$ (as in (Jordon et al., 2019)) and use the implementations' default hyperparameters, with a couple of exceptions. First, we set the maximum number of training iterations to 10,000 to reduce computation. In our experiments, this limit is only reached for `borealis` and `smartnoise` with $\epsilon \geq 10$. Consequently, we train `synthcity` for a set of iterations rather than epochs. Second, for `updated`, we use $\lambda = 0.001$ to prevent the model from spending its privacy budget in just a few iterations. Finally, for all models, we set the number of teachers to $N/1{,}000$ following (Jordon et al., 2019), with the only exception being Kaggle Credit, where we set it to $N/5{,}000$ due to computational constraints; regardless,

---

[7]As per https://pypistats.org/packages/synthcity and https://pypistats.org/packages/smartnoise-synth.

| Classifier | (Jordon et al., 2019) | original | updated | synthcity | turing | borealis | smartnoise |
|---|---|---|---|---|---|---|---|
| Logistic Regression | - | 0.5109 | 0.5198 | 0.5271 | 0.5816 | 0.5304 | 0.5414 |
| Random Forest | - | 0.9529 | 0.7362 | 0.8187 | 0.9574 | 0.8283 | 0.8832 |
| Gaussian Naive Bayes | - | 0.3864 | 0.5385 | 0.9126 | 0.5000 | 0.6515 | 0.6551 |
| Bernoulli Naive Bayes | - | 0.6524 | 0.5185 | 0.9283 | 0.9680 | 0.5141 | 0.6517 |
| Linear SVM | - | 0.5181 | 0.5214 | 0.5333 | 0.5220 | 0.5460 | 0.5384 |
| Decision Tree | - | 0.6359 | 0.6614 | 0.7470 | 0.7003 | 0.6818 | 0.6769 |
| LDA | - | 0.5086 | 0.5229 | 0.5341 | 0.6226 | 0.5420 | 0.5440 |
| AdaBoost | - | 0.9162 | 0.6752 | 0.7588 | 0.9151 | 0.8582 | 0.7572 |
| Bagging | - | 0.8339 | 0.7446 | 0.8285 | 0.8635 | 0.8187 | 0.8386 |
| GBM | - | 0.9376 | 0.6512 | 0.7097 | 0.9188 | 0.7387 | 0.8235 |
| MLP | - | 0.5202 | 0.6194 | 0.7204 | 0.5682 | 0.5439 | 0.6267 |
| XGBoost | - | 0.8678 | 0.7149 | 0.5797 | 0.7927 | 0.8227 | 0.8194 |
| Average | 0.7681 | 0.6867 | 0.6187 | 0.7165 | 0.7425 | 0.6730 | 0.6963 |

Table 5: Performance comparison of 12 classifiers in Setting B (train on synthetic, test on real) in terms of AUROC on UCI Epileptic Seizure ($\epsilon = 1$). Baseline performance: 0.5000. Setting A (train on real, test on real) performance: 0.8103.

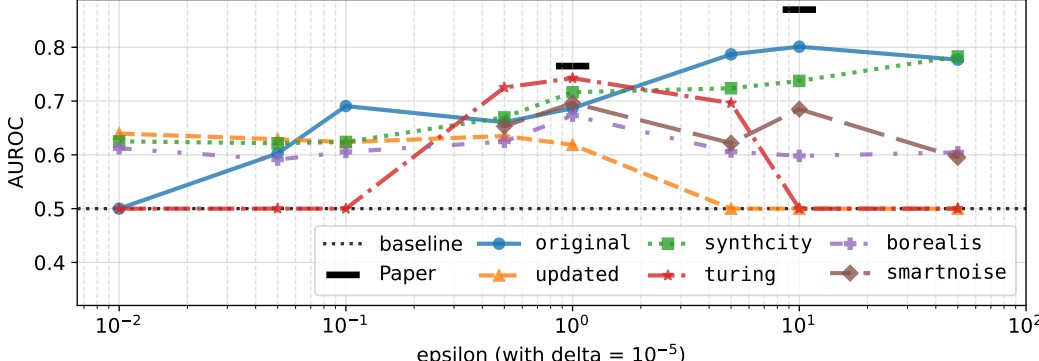

Figure 1: Performance comparison of 12 classifiers (averaged) in Setting B (train on synthetic, test on real) in terms of AUROC with various $\epsilon$ (with $\delta = 10^{-5}$) on UCI Epileptic Seizure.

note that the difference in performance in the original paper (Jordon et al., 2019) is negligible. As done in (Jordon et al., 2019), we also consider the data bounds to be public, i.e., we do not extract the bounds of the data in a DP manner (see Section 4) for `smartnoise`, thus saving its budget for the model training.

**Analysis.** In Table 4, we report the AUROC scores for all datasets and implementations for $\epsilon = 1$. The AUROC scores are averaged over the 12 classifiers (recall that, for each classifier, we consider the best of the scores from the 25 synthetic datasets). For completeness, we also list the scores reported in the paper (Jordon et al., 2019). Due to space limitations, we defer additional/more detailed results to Appendix C – more precisely, in Table 8–9 and Figure 6–7 for Kaggle Credit, Table 10–11 and Figure 8–9 for UCI Epileptic Seizure, and Figure 10 for MNIST. Except for three cases (two of which with `original`), all experiments underperform the results from (Jordon et al., 2019) (by 40.15% on average).

*Kaggle Credit.* Looking closer at Kaggle Credit, which is (Jordon et al., 2019)'s main focus, we note that, out of the implementations that either do not use neural networks as teachers or do not implement PATE correctly (`original`, `updated`, `synthcity`, and `turing`), only `synthcity` performs slightly better than random, i.e., AUROC of 0.5. This is not surprising due to the extreme imbalance of the dataset (only 0.17% of the instances have a positive label) and the known disparate effect of DP (Bagdasaryan et al., 2019; Farrand et al., 2020; Ganev et al., 2022). In other words, even a moderate data imbalance can cause a disproportionate utility drop on the underrepresented class, which is well captured by the AUROC metric. Although `borealis` and `smartnoise` achieve better results (0.6300 and 0.7358, respectively), thus supporting previous claims

| Dataset | Paper (Jordon et al., 2019) | original Non-Conditional | Conditional |
|---|---|---|---|
| Kaggle Credit | 0.8737 | 0.5000 | 0.8760 |
| Kaggle Cervical Cancer | 0.9108 | 0.9265 | 0.9330 |
| UCI ISOLET | 0.6399 | 0.8342 | 0.6927 |
| UCI Epileptic Seizure | 0.7681 | 0.6867 | 0.7029 |
| AUROC $\Delta$ | - | -24.57% | -3.34% |

Table 6: Average AUROC scores of (non-)conditional `original` over the 12 classifiers ($\epsilon = 1$) and reduction in AUROC (0.5 used as baseline) from the original paper (Jordon et al., 2019).

that PATE offers a reduced disparate effect (Uniyal et al., 2021; Ganev, 2022), there is still a considerable gap to the results reported in (Jordon et al., 2019), i.e., 0.8737. After further examining `original` (and the OpenReview discussion (Jordon et al., 2019)), we find that the implementation expects the synthetic data label distribution to be provided, and this is done to exactly match the counts in the training data. We consider this unaccounted privacy leakage, which violates the privacy of the training data, and in our experiments, we do not pass the label distribution.

*UCI Epileptic Seizure.* Next, we focus on the UCI Epileptic Seizure experiments as this: i) has the second most records (11.5k), ii) is high dimensional (179 columns), and iii) has the smallest imbalance (20%). For $\epsilon = 1$, we report the AUROC scores in Table 5, and the scores for $\epsilon$ varying between 0.01 and 50 in Figure 1. Again, none of the implementations come close to the results reported in (Jordon et al., 2019) – note that, for UCI Epileptic Seizure, the authors only report the average scores across the 12 classifiers. Apart from `original`, which does not implement PATE, the only implementation that consistently achieves better utility with increasing $\epsilon$ is `synthcity`. In fact, `borealis` and `smartnoise`'s AUROC peak at $\epsilon = 1$ then slightly drop to around 0.6, while for $\epsilon > 1$, `updated` and `turing`'s drop significantly approaching the random baseline. This is unexpected and could be due to various reasons, e.g., overfitting and mode collapse. Moreover, `synthcity` has an early stopping criterion, which might be another contributing factor. Also note that, even with DP processing disabled, `smartnoise` cannot be trained for $\epsilon < 0.5$. Additionally, in Figure 8 in Appendix C, we plot the mean AUROC scores across the 12 classifiers (rather than the maximum) alongside their standard errors to further confirm that the none of the implementations can reach the results reported in (Jordon et al., 2019) even when accounting for randomness.

*Effect of Number of Teachers.* We also experiment with different numbers of teachers-discriminators $\{N/50, N/100, N/500, N/1,000, N/5,000\}$, similar to the original paper (Jordon et al., 2019), on UCI Epileptic Seizure and present the results in Table 11. While the best AUROC scores for all implementations occur at $N/1,000$, as claimed by (Jordon et al., 2019), not all models behave consistently. Only `synthcity` and `borealis` show improved results as the number of teachers is reduced from $N/50$ to $N/1,000$, with performance decreasing thereafter. By contrast, `updated` and `smartnoise` behave more randomly.

*Effect of Conditional Generation.* As mentioned earlier, unlike (Jordon et al., 2019), we do not use the training labels distribution when generating synthetic data, as this would result in unaccounted privacy leakage. Nevertheless, we compare the performance of non-conditional and conditional generation in Table 6 in an attempt to reach the results in (Jordon et al., 2019). Across all datasets, we observe that conditional generation gets closer to them, most notably for Kaggle Credit, where the AUROC score improves beyond random performance. Overall, the utility drop decreases from -24.57% with non-conditional generation to -3.34% with conditional, effectively reproducing the performance reported in (Jordon et al., 2019).

*Effect of Default Hyperparameters.* For completeness, we also evaluate all implementations using the hyperparameters (and networks depths) specified in the original paper (Jordon et al., 2019), which are listed in the first column of Table 3). The results for all datasets are reported in Table 12 in Appendix C. As expected, the utility of all implementations drops by an average of 17.5% compared to when their respective default hyperparameters are used.

*Comparison with DPGAN.* Finally, we compare the AUROC results of the six implementations to DP-GAN (Xie et al., 2018) as reported in (Jordon et al., 2019) (Kaggle Credit: 0.8578, Kaggle Cervical Cancer:

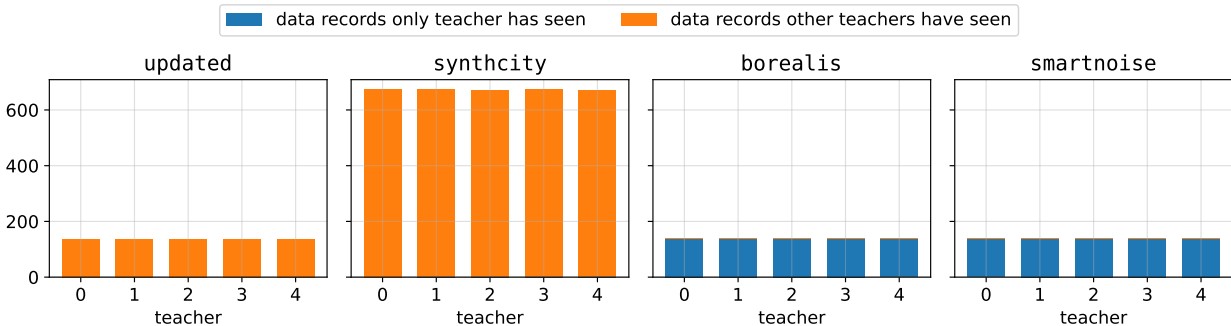

Figure 2: Data records seen by the five teachers-discriminators ($\epsilon = 1$) on Kaggle Cervical Cancer.

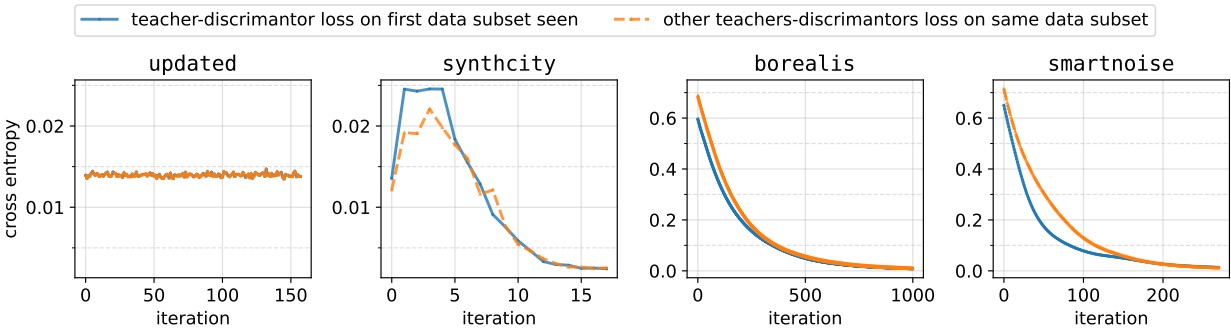

Figure 3: Cross entropy of the five teachers-discriminators on a fixed subset of data ($\epsilon = 1$).

0.8699, UCI ISOLET: 0.5577, UCI Epileptic Seizure: 0.6718). For $\epsilon = 1$, the PATE-GAN implementations perform better in 13 of the 24 experiments, which contradicts the claim in (Jordon et al., 2019) that PATE-GAN is uniformly better than DP-GAN.

# 6 Privacy Evaluation

In this section, we perform an in-depth privacy evaluation of the six PATE-GAN implementations.

## 6.1 PATE-GAN Training

We start by tracking different aspects of the model's training procedure – namely, the records seen by the teachers, the teachers' losses, and the accountant's moments. We do so over a single training run on an average-case dataset, i.e., Kaggle Cervical Cancer. We fix $\epsilon = 1$, except for the moments accountant evaluation, in which we train the models for a fixed 1,000 iterations thus obtaining large $\epsilon$ values. We also set $\delta = 10^{-5}$, $k = 5$, $\lambda = 0.001$, and use all other default hyperparameters.

**Analysis.** First, in Figure 2, we report the number of distinct data points provided as input to the five teachers. We observe that only `borealis` and `smartnoise` correctly partition the data into disjoint subsets and feed them to the corresponding teachers. While `updated` initially separates the data, an indexing bug in the implementation results in all teachers only seeing the last teacher's data (the rest is not used at all). On the other hand, `synthcity` samples records at each iteration, and every teacher ends up seeing all the data. Unfortunately, for `updated` and `synthcity`, this breaks one of the main PATE assumptions (i.e., every teacher can only see a disjoint partition of the data).

Next, in Figure 3, we report the losses of the teachers during training, i.e., the cross entropy of one teacher on its first seen data subset vs. the average of the others on the same subset. Since `updated` and `synthcity` use Logistic Regression instead of neural networks, their initial losses are much lower compared to `borealis`/`smartnoise` (around 30 times lower) as they are (re-)fitted until convergence at every

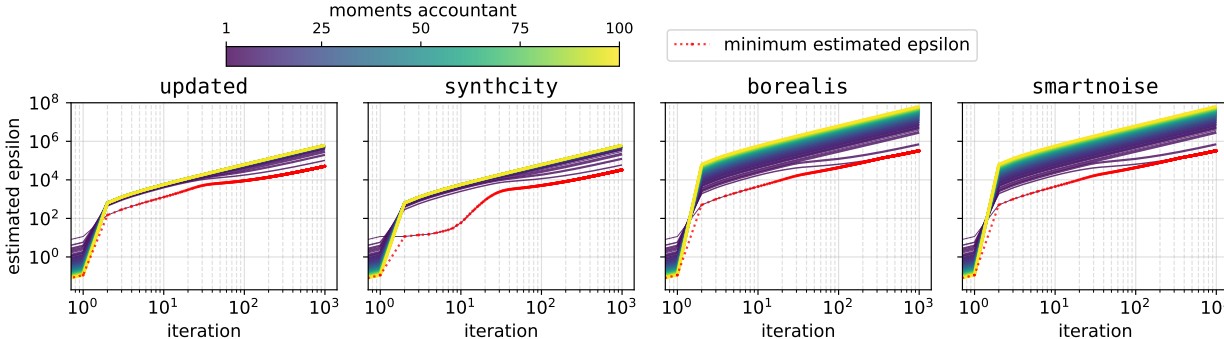

Figure 4: Moments accountant values for 1,000 training iterations.

iteration. Whereas, for `updated`, the two losses are exactly the same since the classifiers are fitted on the same data, the losses of `synthcity` are much "jumpier" because the data changes at each iteration. As for `borealis` and `smartnoise`, their performance is as expected, with the loss on seen data (the blue line in the plot) being initially lower than the one on unseen data (orange line), and both smoothly getting lower with more iterations before converging to approximately the same value. In Figure 11 in Appendix C, we plot the losses over 10 training runs, and observe the same patterns.

Finally, in Figure 4, we plot the 100 moments of the moments accountant over 1,000 iterations. Their scale is very different, with only `borealis` and `smartnoise`'s values being identical. The first moment of `synthcity` (corresponding to the estimated privacy budget) is much lower compared to `updated`, i.e., at iteration 1,000, `synthcity` is 30k vs. 50k for `updated`. After manually inspecting the code, we find an indexing bug for `synthcity`, which, unfortunately, makes it severely underestimate $\epsilon$. By contrast, `borealis` and `smartnoise` massively overestimate $\epsilon$, to around 320k (a multiple of 6 compared to `updated`) after 1,000 iterations. This is due to another bug in the privacy accountant, as a term is not scaled by the log operator. We (re-)list all the violations in Section 6.3.

## 6.2 DP Auditing of PATE-GAN

Our DP auditing procedure uses/adapts two membership inference attacks – namely, GroundHog (Stadler et al., 2022) and Querybased (Houssiau et al., 2022a) – and derives $\epsilon_{emp}$ via the distinguishing game (Annamalai et al., 2024b) discussed in Section 2.

**Adversarial Model.** DP auditing is typically performed in one of two models: black-box, where the MIA adversary only has access to the synthetic data, or white-box, where they can also observe the trained generative model and its internal parameters.[8] For the sake of our experiments, we focus on the former using two black-box attacks (GroundHog and Querybased). We do so as black-box models are considered more realistic to execute in the real world, even though they yield less tight audits (as shown in (Houssiau et al., 2022a; Annamalai et al., 2024b)). In other words, auditing PATE-GAN implementations in the black-box model provides us with a measuring stick for privacy leakage.

GroundHog and Querybased attacks rely on different approaches to featurize, or reduce the dimensionality of, the input synthetic datasets. For the former, we use $F_{naive}$, which extracts every column's min, max, mean, median, and standard deviation. For the latter, we run every possible query, i.e., all values in the dataset's domain (although, in practice, we limit the domain), and count the number of occurrences in the synthetic data. Once the features are extracted, we fit a Random Forest classifier to get $Pred$ and $Pred'$.

**Setup & Hyperparameters.** We run GroundHog on an average-case dataset, i.e., Kaggle Cervical Cancer, choosing a real target from the dataset by running "mini"-MIAs as done in previous work (Meeus et al.,

---

[8]E.g., in a white-box attack vs. GANs like LOGAN (Hayes et al., 2019), the adversary directly leverages the discriminator – trained using DP-SGD and therefore private – to observe its output on the target record (higher confidence usually indicates that the record was part of the training data). However, this approach does not directly apply to PATE-GAN, as the adversary cannot query the individual teachers, which are non-private, but only the student, which is already DP.

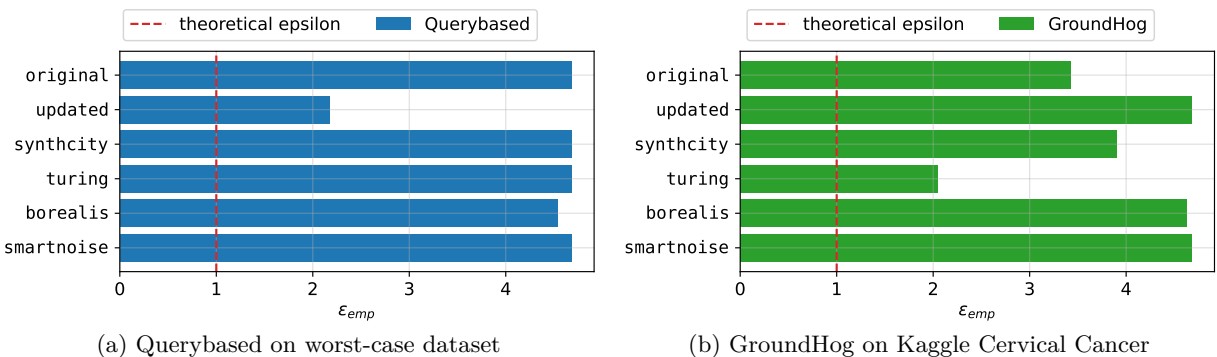

(a) Querybased on worst-case dataset
(b) GroundHog on Kaggle Cervical Cancer

Figure 5: DP auditing with different black-box MIAs ($\epsilon = 1$, as per the dashed red lines).

2023; Annamalai et al., 2024b). This entails running MIAs a limited number of times (we select 100) on a collection of records furthest away from the rest (we choose 64) and selecting the record yielding the highest AUC as the target. For Querybased, we manually craft a worst-case dataset, consisting of 4 repeated (0, 0, 0) records, and a worst-case target record – (1, 1, 1). This limits the number of queries run on the dataset, i.e., all possible combinations, to 8. We run both attacks for 1,000 iterations, using the first 400 outputs to fit the classifier, the next 200 for validation (adjusting the optimal decision boundary), and the final 400 for testing. Finally, note that even if the adversary is 100% correct, the maximum attainable $\epsilon_{emp}$ is around 4.7, a limitation coming from the statistical power of the Clopper-Pearson method (Nasr et al., 2021).

We train all models with $\epsilon = 1$, $\delta = 10^{-5}$, and $\lambda = 0.001$. For Querybased, we use two teachers since there are only four/five records, while for GroundHog we use five teachers. To reduce computation, we set the maximum number of training iterations per generative models to 1,000, as we train 2,000 shadow models per setting (24,000 in total).

**Analysis.** In Figure 5, we plot the empirical privacy estimates ($\epsilon_{emp}$) obtained with the different MIAs. For the Querybased attack (see Figure 5a), we run the MIA on a worst-case dataset and a manually crafted target residing outside the bounds of the data, as previous work (Nasr et al., 2021; 2023; Annamalai et al., 2024b) has shown that these (strong assumptions) are needed to audit DP-SGD and other DP generative models tightly. For all PATE-GAN implementations, we see that $\epsilon_{emp} \gg \epsilon$, which means that we detect privacy violations across all of them.

Next, we relax the adversarial model by running MIAs on an average-case dataset, Kaggle Cervical Cancer, and a real target selected from the data. Running GroundHog results in $\epsilon_{emp} \gg \epsilon$ for all implementations (see Figure 5b). This strongly suggests we are successfully detecting other privacy violations beyond worst-case datasets/targets, which is the main auditing setting used by (Stadler et al., 2022; Annamalai et al., 2024b) against other DP generative models (as discussed in Section 3). Overall, the fact that we obtain $\epsilon_{emp} \gg \epsilon$ (implying large privacy leakage) against PATE-GAN implementations, even in a black-box model, further points to the severity of the privacy violations. The very significant privacy leakage observed in `borealis` and `smartnoise` might seem surprising, but it might be due to algorithmic and privacy analysis differences between PATE-GAN and the original PATE (Papernot et al., 2017), as discussed in Appendix A.2.

We repeat both experiments using the Bayesian credible intervals from (Zanella-Béguelin et al., 2023) (see Figure 12 in Appendix C), and, similarly to the Clopper-Pearson method, we always get $\epsilon_{emp} \gg \epsilon$. We even achieve slightly higher estimates on 9 out of 12 occasions.

**_Remarks._** Overall, our DP auditing procedure follows that of (Annamalai et al., 2024b) and uses elements of (Stadler et al., 2022; Houssiau et al., 2022a; Zanella-Béguelin et al., 2023; Meeus et al., 2023).

Nevertheless, Annamalai et al. (2024b) only audit PrivBayes (Zhang et al., 2017), MST (McKenna et al., 2021), and DP-WGAN (Alzantot & Srivastava, 2019), while we audit PATE-GAN. They show that tight auditing for these models – necessary to effectively identify privacy violations – is only possible with worst-

case datasets/targets and white-box access to the model. Furthermore, while they detect a handful of privacy violations, they do not thoroughly explore why/where these violations happen.

In contrast, we discover numerous privacy violations across all six implementations using only average-case datasets and black-box access to the model (Figure 5b), which are much more conservative and realistic assumptions. Additionally, we provide an in-depth analysis of these privacy violations, including finding the specific lines of code where they occur (Section 6.3).

### 6.3 Summary of Privacy Violations

We now provide an overview of the bugs and privacy violations we discover, also reported in Table 2. When applicable, we highlight them in Algorithms 2, 3, 4, 5, 6, and 7 in Appendix E.

As discussed, `original` and `turing` do not actually implement PATE – no teachers/student (only a single discriminator) and no moments accountant. Both use the Gaussian mechanism (McSherry & Talwar, 2007) instead of Laplace and have errors in the $\delta$ scale, resulting in lower sensitivity. More precisely, `original` uses the XOR operator ($^\wedge$) instead of power (**) in Python, `turing` uses multiplication instead of division. Also, `updated` ingests less noise than necessary as it uses Laplace with standard deviation of $\lambda$ instead of $1/\lambda$. In terms of data partition, neither `original`, `updated`, `synthcity`, nor `turing` separate the data correctly into disjoint sets and/or feed them to the teachers accordingly. Also, `updated` and `synthcity` use Logistic Regression instead of neural networks for the teachers. Moreover, `synthcity`, `borealis`, and `smartnoise` have errors in the moments accountant – the former has an indexing bug while the latter two skip a log operator. At generation, `original` feeds the unperturbed training labels distribution.

Finally, we find that the majority of implementations (`original`, `updated`, `synthcity`, `borealis`) directly extract the data bounds from the data in a non-DP way. Also, `updated` and `borealis` do not return synthetic data in the original scale, while `turing` rounds down all integer values when processing the generated data back to the original bounds.

## 7 Conclusion

This paper presented a benchmark evaluation of six popular implementations of PATE-GAN (Jordon et al., 2019), aiming to 1) reproduce their analysis of the synthetic data's utility on downstream classifiers analysis and 2) perform a deeper privacy analysis. Alas, none of the implementations reproduces the utility reported in (Jordon et al., 2019), achieving, on average, 40% lower AUROC scores. Moreover, our privacy evaluation (including DP auditing) specific to PATE-GAN exposes privacy violations and bugs in all six implementations (cf. Table 2).

Numerous privacy-preserving technologies, including DP synthetic data, are already deployed in critical industries such as healthcare and finance (NHS, 2021; Giuffrè & Shung, 2023; FCA, 2024). Ensuring robust privacy protection in these applications is essential, as failures could lead to catastrophic breaches and the exposure of individuals' sensitive data. However, implementing DP mechanisms correctly in practice is highly complex, as many privacy-related bugs can be easily overlooked, or worse, challenging or even impossible to detect through manual code review. Papers like ours help automate the testing and verification of DP implementations' theoretical protections, or uncover discrepancies. Our auditing scheme can serve as an inspiration for researchers and developers to validate their own DP implementations. As a result, we believe our work will encourage more reproducible research and greater trust in the field.

**Acknowledgements.** We are grateful to the TMLR Action Editor and reviewers for their valuable feedback and suggestions, which helped us to significantly improve our paper. We also thank Bogdan Kulynych for pointing us to discrepancies between PATE-GAN and PATE.

**Responsible Disclosure.** In the spirit of responsible disclosure, in June 2024, we contacted the authors of the six implementations, sharing the detected DP violations via emails and GitHub issues, along with the exact lines of code where they occur and potential fixes. More specifically, we sent emails to the authors of `original` and `turing` and raised 11 GitHub issues (see Table 13 in Appendix D). As of February 2025, none of the bugs have been fixed.

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

# A PATE-GAN Algorithm and Differences with PATE

## A.1 PATE-GAN Algorithm

Algorithm 1 reports the original PATE-GAN training procedure (Jordon et al., 2019) discussed in Section 2.

---

**Algorithm 1** Pseudo-code of PATE-GAN

---

1: **Input:** $\delta, \mathcal{D}, n_T, n_S$, batch size $n$, number of teachers $k$, noise size $\lambda$
2: **Initialize:** $\theta_G, \theta_T^1, \ldots, \theta_T^k, \theta_S, \alpha(l) = 0$ for $l = 1, \ldots, L$
3: Partition dataset into $k$ subsets $\mathcal{D}_1, \ldots, \mathcal{D}_k$ of size $\frac{|\mathcal{D}|}{k}$
4: **while** $\hat{\epsilon} < \epsilon$ **do**
5:     **for** $t_2 = 1, \ldots, n_T$ **do**
6:         Sample $\mathbf{z}_1, \ldots, \mathbf{z}_n \overset{\text{i.i.d.}}{\sim} P_{\mathcal{Z}}$
7:         **for** $i = 1, \ldots, k$ **do**
8:             Sample $\mathbf{u}_1, \ldots, \mathbf{u}_n \overset{\text{i.i.d.}}{\sim} \mathcal{D}_i$
9:             Update teacher, $T_i$, using SGD
10:             $\nabla_{\theta_T^i} - \left[ \sum_{j=1}^d \log(T_i(\mathbf{u}_j)) + \log(1 - T_i(G(\mathbf{z}_j))) \right]$
11:         **end for**
12:     **end for**
13:     **for** $t_3 = 1, \ldots, n_S$ **do**
14:         Sample $\mathbf{z}_1, \ldots, \mathbf{z}_n \overset{\text{i.i.d.}}{\sim} P_{\mathcal{Z}}$
15:         **for** $j = 1, \ldots, n$ **do**
16:             $\hat{\mathbf{u}}_j \leftarrow G(\mathbf{z}_j)$
17:             $r_j \leftarrow \text{PATE}_\lambda(\hat{\mathbf{u}}_i)$ for $j = 1, \ldots, n$
18:             Update moments accountant
19:             $q \leftarrow \frac{2 + \lambda |n_0 - n_1|}{4 \exp(\lambda |n_0 - n_1|)}$
20:             **for** $l = 1, \ldots, L$ **do**
21:                 $\alpha(l) \leftarrow \alpha(l) + \min\{2\lambda^2 l(l+1), \log((1-q)\left(\frac{1-q}{1-e^{2\lambda}q}\right)^l + qe^{2\lambda l})\}$
22:             **end for**
23:             Update the student, $S$, using SGD
24:             $\nabla_{\theta_S} - \sum_{j=1}^n r_j \log S(\hat{\mathbf{u}}_j) + (1 - r_j) \log(1 - S(\hat{\mathbf{u}}_j))$
25:         **end for**
26:     Sample $\mathbf{z}_1, \ldots, \mathbf{z}_n \overset{\text{i.i.d.}}{\sim} P_{\mathcal{Z}}$
27:     Update the generator, $G$, using SGD
28:     $\nabla_{\theta_G} \left[ \sum_{i=1}^n \log(1 - S(G(\mathbf{z}_j))) \right]$
29:     $\hat{\epsilon} \leftarrow \min_l \frac{\alpha(l) + \log(\frac{1}{\delta})}{l}$
30:     **end for**
31: **end while**
32: **Output:** $G$

---

## A.2 Differences between PATE-GAN and PATE

We now highlight key differences between the privacy analyses of PATE-GAN (Jordon et al., 2019) and PATE (Papernot et al., 2017) (recall that the former is directly based on the latter).

First, the notation used by (Jordon et al., 2019) and (Papernot et al., 2017) are inconsistent with each other. Specifically, in Section 3.2 (Jordon et al., 2019) define the PATE$_{\boldsymbol{\lambda}}$ mechanism to add noise $Lap(\boldsymbol{\lambda})$ and satisfy $(\frac{\mathbf{1}}{\boldsymbol{\lambda}}, 0)$-DP. However, in Theorem 2, (Papernot et al., 2017) originally define the PATE$_{\boldsymbol{\gamma}}$ mechanism to add noise $Lap(\frac{\mathbf{1}}{\boldsymbol{\gamma}})$ and satisfy $(\mathbf{2}\boldsymbol{\gamma}, 0)$-DP.[9] This has two consequences: i) it is unclear why PATE-GAN's privacy analysis is different by a factor of 2, and ii) the noise parameter $\boldsymbol{\lambda}$ in PATE-GAN is inversely related

---

[9]Notation is bolded here for emphasis but in the original papers appear unbolded.

to $\gamma$ in PATE. However, the privacy analysis of PATE-GAN – specifically, the moments accountant (lines 18-21 in Algorithm 1) and Theorem 5 in (Jordon et al., 2019), both based on Theorem 3 and Lemma 4 in (Papernot et al., 2017) – directly substitutes $\lambda$ for $\gamma$ (not $\frac{1}{\gamma}$) and includes the factor 2. Moreover, `updated` and `synthcity`, the two implementations from the original authors implementing PATE, are inconsistent with each other. When aggregating the teachers' votes, the former adds noise $Lap(\lambda)$, while the latter $Lap(\frac{1}{\lambda})$. Then, both update their accountants according to Algorithm 1 (i.e., using $\lambda$). Overall, while the notations in PATE-GAN (Jordon et al., 2019)/`updated`/`synthcity` might be inconsistent, `synthcity` seems to follow PATE (Papernot et al., 2017) the closest (but still omits the factor 2).

Next, we examine the moments accountant updates (line 21 in Algorithm 1). Theorem 5 in (Jordon et al., 2019) overlooks a condition for q, namely, $q < \frac{e^{2\gamma}-1}{e^{4\gamma}-1}$, as stated by Theorem 3 in (Papernot et al., 2017). Additionally, PATE-GAN's accountant update takes the minimum over two factors, whereas PATE's implementation includes a third one, $\gamma l$.[10]

Finally, PATE (Papernot et al., 2017) explains that since the privacy budget analysis is data-dependent, $\epsilon$ should itself be released in a DP way. Their implementation follows this principle,[10] whereas PATE-GAN overlooks it and outputs the unperturbed $\epsilon$.

Overall, during our utility and privacy experiments, we follow PATE-GAN's analysis from (Jordon et al., 2019) and leave the analysis and resolution of these discrepancies with PATE (Papernot et al., 2017) for future work.

## B  Additional Preliminaries

### B.1  Datasets

We use four of the original six datasets from (Jordon et al., 2019) as the other two are not publicly available – specifically, Kaggle Credit (Pozzolo et al., 2015), Kaggle Cervical Cancer (Fernandes et al., 2017), UCI ISOLET (Cole & Fanty, 1994), UCI Epileptic Seizure (Andrzejak et al., 2001). We also use MNIST (LeCun et al., 2010), the popular digits dataset. The main characteristics of these datasets are summarized in Table 7. When needed, we follow (Jordon et al., 2019) to convert the downstream tasks associated with the datasets into binary classification.

Kaggle Credit is a dataset consisting of 284,807 credit card transactions labeled as fraudulent or not. Besides the labels, there are 29 numerical features. The dataset is highly imbalanced, only 492 transactions (0.17%) are actually fraudulent.

Kaggle Cervical Cancer contains the demographic information and medical history of 858 patients. There are 35 features, 24 binary/11 numerical, and a biopsy status label. Only 55 patients (6.4%) have positive biopsies.

The UCI ISOLET dataset has 7,797 featurized pronunciations (617 numerical features) of a letter of the alphabet (the label). We transform the task to classifying vowels vs. consonants. Out of the 7,797 letters, there are 1,500 (19.2%) vowels.

UCI Epileptic Seizure includes the brain activities encoded into 178 numerical vectors of 11,500 patients. Originally, there were five distinct labels, which we transformed into a binary one to indicate whether there was a seizure activity. This results in 2,300 (20%) records with a positive label.

MNIST is a benchmark image dataset consisting of 60,000 grayscale handwritten digits, each of size $28\times28$ pixels. The associated task is digit classification. This is the most balanced dataset, with the ratio of the least to the most frequent digit being 80.4%.

---

[10]As per: https://github.com/tensorflow/privacy/blob/master/research/pate_2017/analysis.py.

| Dataset | $N$ | $d$ | Imbalance | AUROC | AUPRC |
|---|---|---|---|---|---|
| Kaggle Credit | 284,807 | 30 | 0.0017 | 0.8176 | 0.5475 |
| Kaggle Cervical Cancer | 858 | 36 | 0.0641 | 0.9400 | 0.6192 |
| UCI ISOLET | 7,797 | 618 | 0.1924 | 0.9678 | 0.9002 |
| UCI Epileptic Seizure | 11,500 | 179 | 0.2000 | 0.8103 | 0.7403 |
| MNIST | 60,000 | 785 | 0.8042 | 0.9960 | 0.9792 |

Table 7: Summary of the datasets used in our evaluations.

## B.2 Classifiers and Evaluation Metrics

Following the original PATE-GAN paper (Jordon et al., 2019), we use the same 12 predictive models to evaluate PATE-GAN's utility performance. Eleven of them are from the popular Python library scikit-learn (Pedregosa et al., 2011); we list them with the names of the algorithms as found in the library in brackets – Logistic Regression (*LogisticRegression*), Random Forest (*RandomForestClassifier*), Gaussian Naive Bayes (*GaussianNB*), Bernoulli Naive Bayes (*BernoulliNB*), Linear Support Vector Machine (*LinearSVC*), Decision Tree (*DecisionTreeClassifier*), Linear Discriminant Analysis Classifier (*LinearDiscriminantAnalysis*), Adaptive Boosting (*AdaBoostClassifier*), Bootstrap Aggregating (*BaggingClassifier*), Gradient Boosting Machine (*GradientBoostingClassifier*), Multi-layer Perceptron (*MLPClassifier*). The twelfth model is XGBoost (*XGBRegressor*) from the library xgboost (Chen & Guestrin, 2016).

As done in (Jordon et al., 2019), we report the Area Under the Receiver Operating Characteristic curve (AUROC) and the Area Under the Precision-Recall Curve (AUPRC) scores to quantify their performance on the classification task.

Finally, all experiments are run on an AWS instance (m4.4xlarge) with a 2.4GHz Intel Xeon E5-2676 v3 (Haswell) processor, 16 vCPUs, and 64GB RAM.

## C Additional Experimental Results

In this Appendix, we present the AUROC and AUPRC scores for Kaggle Credit in Table 8–9 and Figure 6–7, and for UCI Epileptic Seizure in Table 10 and Figure 8–9. We also show the effect of number of teachers-discriminators and default hyperparameters in Table 11 and 12, respectively. We discuss them in Section 5. Additionally, we present further privacy results in Figure 11 and 12, which are discussed in Section 6.

*Kaggle Credit.* On Kaggle Credit, in Setting A (train on real, test on real), we get a lower AUROC score (0.8176) compared to both the original paper's (Jordon et al., 2019) Setting A (0.9438) and Setting B (0.8737) scores. This could be due to two reasons: 1) we set aside 20% of the data for testing, while the `updated`'s authors do it with around 50% and 2) we fit the data preprocessor (min-max scaling) on the training data only; while they do on the combined training and test.

*MNIST.* Finally, we evaluate utility on MNIST. Unlike the four datasets studied thus far, MNIST is i) balanced, with all labels approximately equally represented, and ii) more complex, requiring high-level correlations typically present in image data to be captured by the generative model. In Figure 10, we report the AUROC scores of the six implementations at varying $\epsilon$ levels.

In contrast to the performance observed on the other datasets, there is a distinct difference between `original`, `turing` and the remaining implementations. The first two achieve significantly higher utility, which is expected, as they do not implement PATE correctly. Consistent with prior observations, `synthcity`, shows slight improvement with higher $\epsilon$ values, while `updated`, `borealis`, and `smartnoise` experience performance drops between $\epsilon = 1$ and $\epsilon = 10$. Overall, none of the six implementations demonstrate promising results on MNIST.

| | (Jordon et al., 2019) | original | updated | synthcity | turing | borealis | smartnoise |
|---|---|---|---|---|---|---|---|
| Logistic Regression | 0.8728 | 0.5000 | 0.5000 | 0.2997 | 0.5000 | 0.8720 | 0.8158 |
| Random Forest | 0.8980 | 0.5000 | 0.5000 | 0.7809 | 0.5000 | 0.5076 | 0.7980 |
| Gaussian Naive Bayes | 0.8817 | 0.5000 | 0.5000 | 0.5000 | 0.5000 | 0.5000 | 0.5000 |
| Bernoulli Naive Bayes | 0.8968 | 0.5000 | 0.5000 | 0.6392 | 0.5000 | 0.5480 | 0.6003 |
| Linear SVM | 0.7523 | 0.5000 | 0.5000 | 0.2599 | 0.5000 | 0.8622 | 0.8168 |
| Decision Tree | 0.9011 | 0.5000 | 0.5000 | 0.6068 | 0.5000 | 0.5721 | 0.7013 |
| LDA | 0.8510 | 0.5000 | 0.5000 | 0.2596 | 0.5000 | 0.5000 | 0.8192 |
| AdaBoost | 0.8952 | 0.5000 | 0.5000 | 0.5588 | 0.5000 | 0.5975 | 0.7749 |
| Bagging | 0.8877 | 0.5000 | 0.5000 | 0.7006 | 0.5000 | 0.5051 | 0.7688 |
| GBM | 0.8709 | 0.5000 | 0.5000 | 0.5342 | 0.5000 | 0.5928 | 0.6987 |
| MLP | 0.8925 | 0.5000 | 0.5000 | 0.3242 | 0.5000 | 0.8822 | 0.8144 |
| XGBoost | 0.8904 | 0.5000 | 0.5000 | 0.7426 | 0.5000 | 0.6202 | 0.7213 |
| Average | 0.8737 | 0.5000 | 0.5000 | 0.5172 | 0.5000 | 0.6300 | 0.7358 |

Table 8: Performance comparison of 12 classifiers in Setting B (train on synthetic, test on real) in terms of AUROC on Kaggle Credit ($\epsilon = 1$). Baseline performance: 0.5000. Setting A (train on real, test on real) performance: 0.8176.

| | (Jordon et al., 2019) | original | updated | synthcity | turing | borealis | smartnoise |
|---|---|---|---|---|---|---|---|
| Logistic Regression | 0.3907 | 0.0017 | 0.0017 | 0.0013 | 0.0017 | 0.4847 | 0.2401 |
| Random Forest | 0.3157 | 0.0017 | 0.0017 | 0.0493 | 0.0017 | 0.0701 | 0.0370 |
| Gaussian Naive Bayes | 0.1858 | 0.0017 | 0.0017 | 0.0017 | 0.0017 | 0.0219 | 0.0017 |
| Bernoulli Naive Bayes | 0.2099 | 0.0017 | 0.0017 | 0.0359 | 0.0017 | 0.4027 | 0.1506 |
| Linear SVM | 0.4466 | 0.0017 | 0.0017 | 0.0012 | 0.0017 | 0.4027 | 0.2506 |
| Decision Tree | 0.3978 | 0.0017 | 0.0017 | 0.0023 | 0.0017 | 0.0023 | 0.0055 |
| LDA | 0.1852 | 0.0017 | 0.0017 | 0.0012 | 0.0017 | 0.0017 | 0.2699 |
| AdaBoost | 0.4366 | 0.0017 | 0.0017 | 0.1678 | 0.0017 | 0.2073 | 0.0927 |
| Bagging | 0.3221 | 0.0017 | 0.0017 | 0.0051 | 0.0017 | 0.0119 | 0.0564 |
| GBM | 0.2974 | 0.0017 | 0.0017 | 0.0572 | 0.0017 | 0.1866 | 0.0222 |
| MLP | 0.4693 | 0.0017 | 0.0017 | 0.0020 | 0.0017 | 0.4453 | 0.1840 |
| XGBoost | 0.3700 | 0.0017 | 0.0017 | 0.0590 | 0.0017 | 0.1254 | 0.0986 |
| Average | 0.3351 | 0.0017 | 0.0017 | 0.0320 | 0.0017 | 0.1635 | 0.1174 |

Table 9: Performance comparison of 12 classifiers in Setting B (train on synthetic, test on real) in terms of AUPRC on Kaggle Credit ($\epsilon = 1$). Baseline performance: 0.0017. Setting A (train on real, test on real) performance: 0.5475.

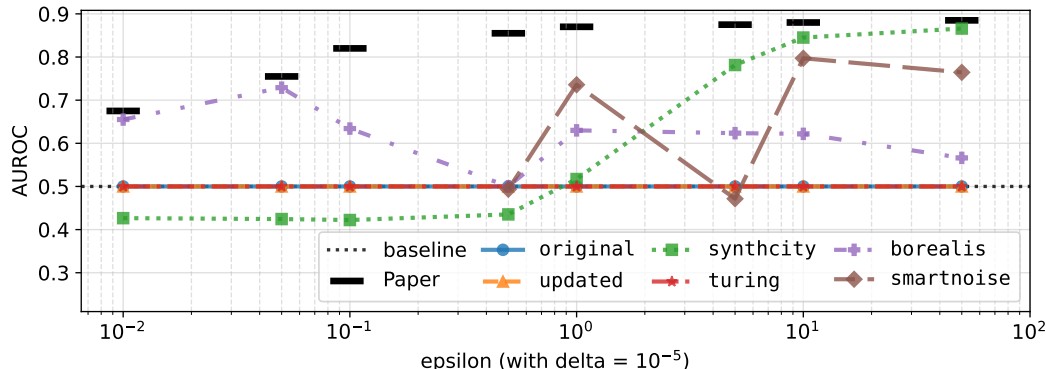

Figure 6: Performance comparison of 12 classifiers (averaged) in Setting B (train on synthetic, test on real) in terms of AUROC with various $\epsilon$ (with $\delta = 10^{-5}$) on Kaggle Credit.

|  | (Jordon et al., 2019) | original | updated | synthcity | turing | borealis | smartnoise |
|---|---|---|---|---|---|---|---|
| Logistic Regression | - | 0.4329 | 0.4717 | 0.4800 | 0.4543 | 0.4765 | 0.5152 |
| Random Forest | - | 0.8902 | 0.5754 | 0.6157 | 0.8789 | 0.6936 | 0.7572 |
| Gaussian Naive Bayes | - | 0.2568 | 0.4938 | 0.8621 | 0.2000 | 0.4338 | 0.5881 |
| Bernoulli Naive Bayes | - | 0.4435 | 0.2409 | 0.7970 | 0.8982 | 0.2282 | 0.4415 |
| Linear SVM | - | 0.4872 | 0.4713 | 0.4843 | 0.4922 | 0.5060 | 0.5128 |
| Decision Tree | - | 0.4045 | 0.4067 | 0.5138 | 0.4371 | 0.4304 | 0.3759 |
| LDA | - | 0.4676 | 0.4725 | 0.4835 | 0.3695 | 0.4990 | 0.4861 |
| AdaBoost | - | 0.7766 | 0.5808 | 0.4836 | 0.8035 | 0.7116 | 0.6086 |
| Bagging | - | 0.7213 | 0.5172 | 0.5548 | 0.6946 | 0.6269 | 0.7092 |
| GBM | - | 0.8563 | 0.5224 | 0.4861 | 0.7030 | 0.5548 | 0.6892 |
| MLP | - | 0.4780 | 0.5739 | 0.6487 | 0.4375 | 0.4867 | 0.5601 |
| XGBoost | - | 0.8094 | 0.5442 | 0.3615 | 0.6587 | 0.7136 | 0.6846 |
| Average | 0.6512 | 0.5854 | 0.4892 | 0.5643 | 0.5856 | 0.5301 | 0.5774 |

Table 10: Performance comparison of 12 classifiers in Setting B (train on synthetic, test on real) in terms of AUPRC on UCI Epileptic Seizure ($\epsilon = 1$). Baseline performance: 0.2000. Setting A (train on real, test on real) performance: 0.7403.

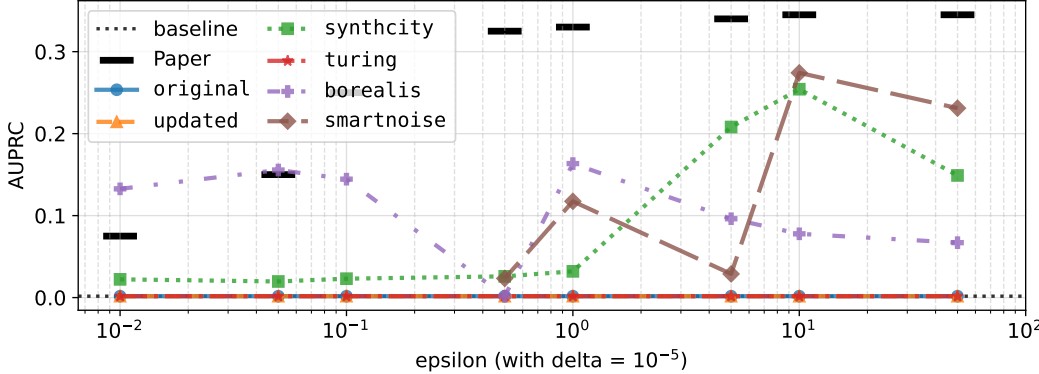

Figure 7: Performance comparison of 12 classifiers (averaged) in Setting B (train on synthetic, test on real) in terms of AUPRC with various $\epsilon$ (with $\delta = 10^{-5}$) on Kaggle Credit.

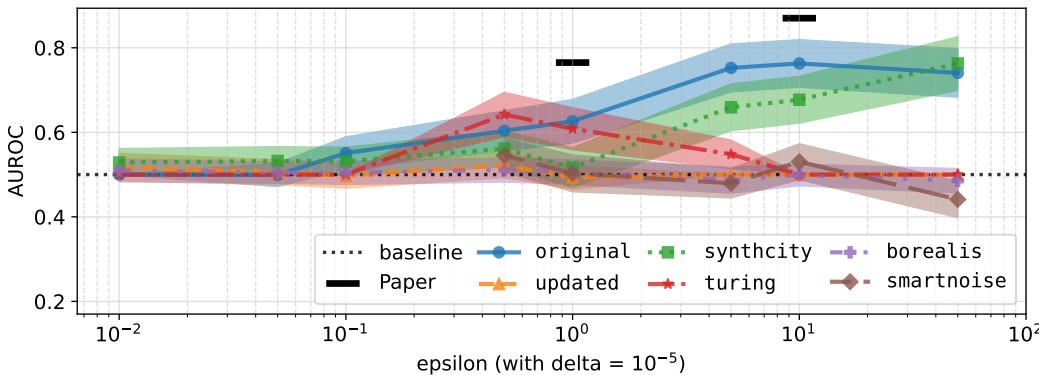

Figure 8: Performance comparison of 12 classifiers (averaged) in Setting B (train on synthetic, test on real) in terms of mean (not maximum) and standard error AUROC with various $\epsilon$ (with $\delta = 10^{-5}$) on UCI Epileptic Seizure.

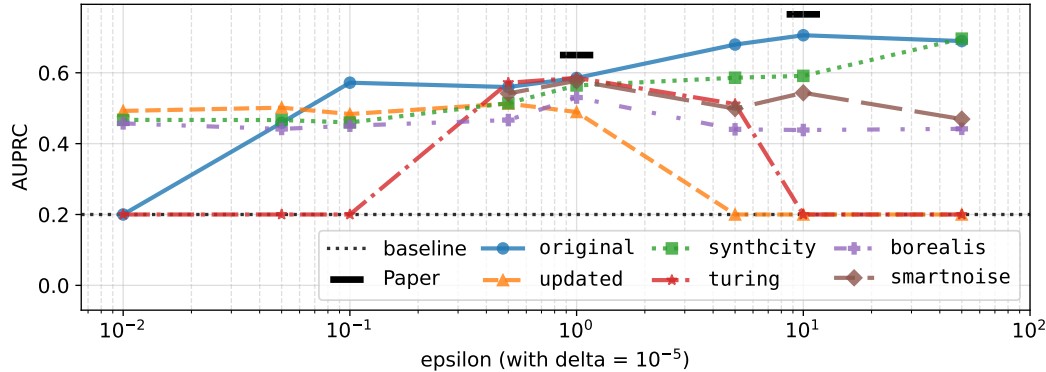

Figure 9: Performance comparison of 12 classifiers (averaged) in Setting B (train on synthetic, test on real) in terms of AUPRC with various $\epsilon$ (with $\delta = 10^{-5}$) on UCI Epileptic Seizure.

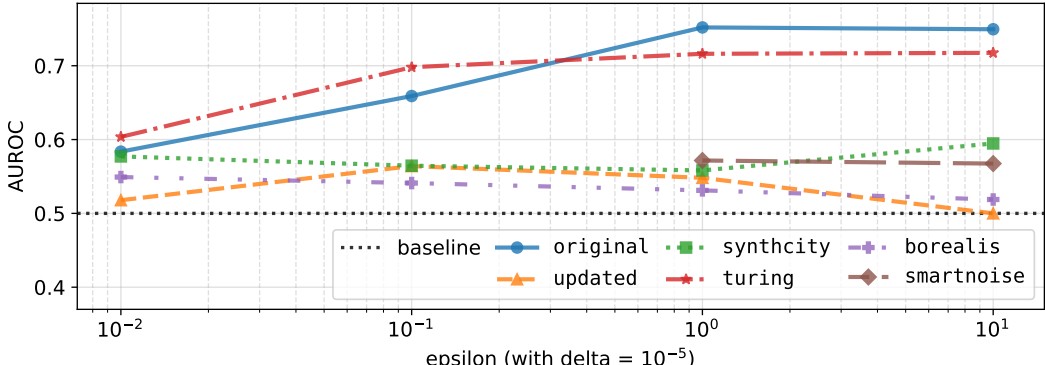

Figure 10: Performance comparison of 2 classifiers (Logistic Regression and Random Forest, averaged) in Setting B (train on synthetic, test on real) in terms of AUROC with various $\epsilon$ (with $\delta = 10^{-5}$) on MNIST.

| Implementation | $N/50$ | $N/100$ | $N/500$ | $N/1,000$ | $N/5,000$ |
|---|---|---|---|---|---|
| updated | 0.6170 | 0.5944 | 0.6077 | **0.6187** | 0.6180 |
| synthcity | 0.6230 | 0.6447 | 0.6643 | **0.7165** | 0.6428 |
| borealis | 0.6170 | 0.6230 | 0.6332 | **0.6730** | 0.6363 |
| smartnoise | 0.6625 | 0.6275 | 0.6653 | **0.6963** | 0.6270 |

Table 11: Tradeoff between number of teachers and performances of 12 classifiers (averaged) in Setting B (train on synthetic, test on real) in terms of AUROC (with $\epsilon = 1$ and $\delta = 10^{-5}$) on UCI Epileptic Seizure.

| | original | | updated | | synthcity | | turing | | borealis | | smartnoise | |
|---|---|---|---|---|---|---|---|---|---|---|---|---|
| Dataset | D | P | D | P | D | P | D | P | D | P | D | P |
| Kaggle Credit | 0.5000 | 0.5000 | 0.5000 | 0.5000 | 0.5172 | 0.5437 | 0.5000 | 0.5000 | 0.6300 | 0.6616 | 0.7358 | 0.5867 |
| Kaggle Cervical | 0.9265 | 0.8531 | 0.8739 | 0.6581 | 0.8584 | 0.6867 | 0.5000 | 0.5000 | 0.9431 | 0.8140 | 0.8025 | 0.7699 |
| UCI ISOLET | 0.8342 | 0.8355 | 0.5861 | 0.6001 | 0.6292 | 0.6091 | 0.5000 | 0.5000 | 0.6219 | 0.5980 | 0.6152 | 0.6009 |
| UCI Epileptic | 0.6867 | 0.6876 | 0.6187 | 0.6223 | 0.7165 | 0.6888 | 0.7425 | 0.5000 | 0.6730 | 0.6069 | 0.6963 | 0.6768 |
| AUROC $\Delta$ | - | -4.23% | - | -13.69% | - | -17.78% | - | -25.00% | - | -20.49% | - | -24.10% |

Table 12: Average AUROC scores of using the default hyperparameters (denoted as **D**) per implementation and the hyperparameters specified in the paper (Jordon et al., 2019) (**P**) over the 12 classifiers ($\epsilon = 1$) as well as reduction in AUROC from **D** to **P**.

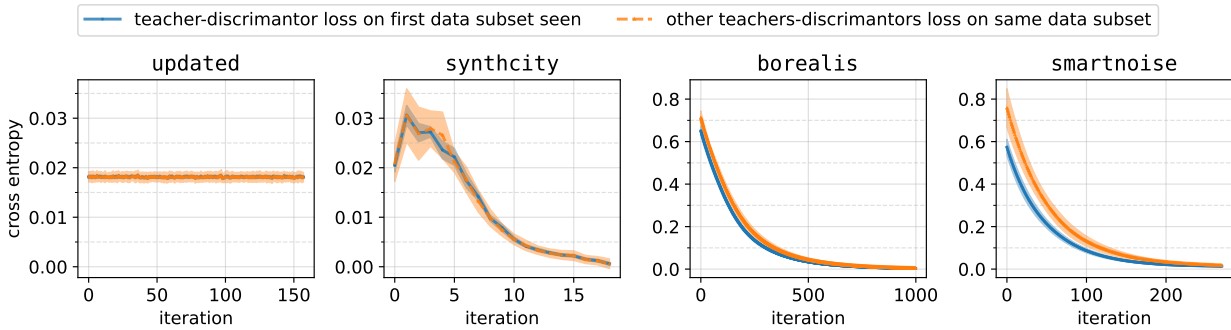

Figure 11: Cross entropy of the five teachers-discriminators on a fixed subset of data ($\epsilon = 1$) over 10 runs.

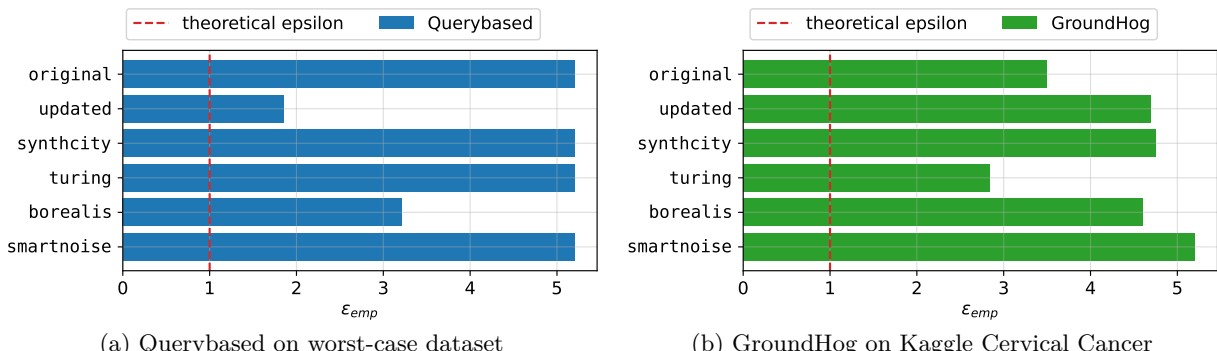

(a) Querybased on worst-case dataset

(b) GroundHog on Kaggle Cervical Cancer

Figure 12: DP auditing using Bayesian credible intervals (Zanella-Béguelin et al., 2023) with different black-box MIAs ($\epsilon = 1$, as per the dashed red lines).

## D   Responsible Disclosure

As mentioned in Section 7, we sent emails to the authors of `original` and `turing`, detailing our privacy concerns, and raised 11 GitHub issues, as detailed in Table 13.

| Implementation | Privacy Violation/Bug | GitHub Issue Link |
|---|---|---|
| updated | Data Partition | https://github.com/vanderschaarlab/mlforhealthlabpub/issues/29 |
| updated | Teachers | https://github.com/vanderschaarlab/mlforhealthlabpub/issues/30 |
| updated | Processing & Metadata | https://github.com/vanderschaarlab/mlforhealthlabpub/issues/31 |
| updated | Laplace Noise | https://github.com/vanderschaarlab/mlforhealthlabpub/issues/36 |
| synthcity | Data Partition | https://github.com/vanderschaarlab/synthcity/issues/275 |
| synthcity | Moments Accountant | https://github.com/vanderschaarlab/synthcity/issues/276 |
| synthcity | Teachers | https://github.com/vanderschaarlab/synthcity/issues/277 |
| synthcity | Metadata | https://github.com/vanderschaarlab/synthcity/issues/278 |
| borealis | Moments Accountant | https://github.com/BorealisAI/private-data-generation/issues/11 |
| borealis | Processing & Metadata | https://github.com/BorealisAI/private-data-generation/issues/12 |
| smartnoise | Moments Accountant | https://github.com/opendp/smartnoise-sdk/issues/596 |

Table 13: Summary of the GitHub issues raised and their corresponding privacy violation or bug.

## E   PATE-GAN Implementations

In Algorithms 2, 3, 4, 5, 6, and 7 we highlight the deviations between the six implementations and the original paper's Algorithm 1 (Jordon et al., 2019).

**Algorithm 2** Pseudo-code of PATE-GAN; `original`

---

1: **Input:** $\delta, \mathcal{D}, n_T, n_S$, batch size $n$, number of teachers $k$, noise size $\lambda$
2: **Initialize:** $\theta_G, \theta_T^1, \ldots, \theta_T^k, \theta_S, \alpha(l) = 0$ **for** $l = 1, \ldots, L$
3: **Partition dataset into $k$ subsets $\mathcal{D}_1, \ldots, \mathcal{D}_k$ of size $\frac{|\mathcal{D}|}{k}$**
4: **while** $\hat{\epsilon} < \epsilon$ **do**
5:      **for** $t_2 = 1, \ldots, n_T$ **do**
6:          Sample $\mathbf{z}_1, \ldots, \mathbf{z}_n \overset{\text{i.i.d.}}{\sim} P_{\mathcal{Z}}$
7:          **for** $i = 1, \ldots, k$ **do**
8:              **Sample $\mathbf{u}_1, \ldots, \mathbf{u}_n \overset{\text{i.i.d.}}{\sim} \mathcal{D}_i$**
9:              **Update teacher, $T_i$, using SGD**
10:              $\nabla_{\theta_T^i} - \left[ \sum_{j=1}^d \log(T_i(\mathbf{u}_j)) + \log(1 - T_i(G(\mathbf{z}_j))) \right]$
11:          **end for**
12:      **end for**
13:      **for** $t_3 = 1, \ldots, n_S$ **do**
14:          Sample $\mathbf{z}_1, \ldots, \mathbf{z}_n \overset{\text{i.i.d.}}{\sim} P_{\mathcal{Z}}$
15:          **for** $j = 1, \ldots, n$ **do**
16:              $\hat{\mathbf{u}}_j \leftarrow G(\mathbf{z}_j)$
17:              $r_j \leftarrow \mathbf{PATE}_\lambda(\hat{\mathbf{u}}_i)$ **for** $j = 1, \ldots, n$
18:              **Update moments accountant**
19:              $q \leftarrow \frac{2 + \lambda|n_0 - n_1|}{4 \exp(\lambda|n_0 - n_1|)}$
20:              **for** $l = 1, \ldots, L$ **do**
21:                  $\alpha(l) \leftarrow \alpha(l) + \min\{2\lambda^2 l(l+1), \log((1-q)\left(\frac{1-q}{1 - e^{2\lambda}q}\right)^l + q e^{2\lambda l})\}$
22:              **end for**
23:          Update the student, $S$, using SGD
24:          $\nabla_{\theta_S} - \sum_{j=1}^n r_j \log S(\hat{\mathbf{u}}_j) + (1 - r_j) \log(1 - S(\hat{\mathbf{u}}_j))$
25:          **end for**
26:      Sample $\mathbf{z}_1, \ldots, \mathbf{z}_n \overset{\text{i.i.d.}}{\sim} P_{\mathcal{Z}}$
27:      Update the generator, $G$, using SGD
28:      $\nabla_{\theta_G} \left[ \sum_{i=1}^n \log(1 - S(G(\mathbf{z}_j))) \right]$
29:      $\hat{\epsilon} \leftarrow \min_l \frac{\alpha(l) + \log(\frac{1}{\delta})}{l}$
30:      **end for**
31: **end while**
32: **Output:** $G$

---

---

**Algorithm 3** Pseudo-code of PATE-GAN; updated

---

1: **Input:** $\delta, \mathcal{D}, n_T, n_S$, batch size $n$, number of teachers $k$, noise size $\lambda$
2: **Initialize:** $\theta_G, \theta_T^1, \ldots, \theta_T^k, \theta_S, \alpha(l) = 0$ for $l = 1, \ldots, L$
3: Partition dataset into $k$ subsets $\mathcal{D}_1, \ldots, \mathcal{D}_k$ of size $\frac{|\mathcal{D}|}{k}$
4: **while** $\hat{\epsilon} < \epsilon$ **do**
5:    **for** $t_2 = 1, \ldots, n_T$ **do**
6:       Sample $\mathbf{z}_1, \ldots, \mathbf{z}_n \overset{\text{i.i.d.}}{\sim} P_{\mathcal{Z}}$
7:       **for** $i = 1, \ldots, k$ **do**
8:          **Sample $\mathbf{u}_1, \ldots, \mathbf{u}_n \overset{\text{i.i.d.}}{\sim} \mathcal{D}_i$**
9:          **Update teacher, $T_i$, using SGD**
10:          $\nabla_{\theta_T^i} - \left[ \sum_{j=1}^{d} \log(T_i(\mathbf{u}_j)) + \log(1 - T_i(G(\mathbf{z}_j))) \right]$
11:       **end for**
12:    **end for**
13:    **for** $t_3 = 1, \ldots, n_S$ **do**
14:       Sample $\mathbf{z}_1, \ldots, \mathbf{z}_n \overset{\text{i.i.d.}}{\sim} P_{\mathcal{Z}}$
15:       **for** $j = 1, \ldots, n$ **do**
16:          $\hat{\mathbf{u}}_j \leftarrow G(\mathbf{z}_j)$
17:          $r_j \leftarrow \text{PATE}_\lambda(\hat{\mathbf{u}}_i)$ for $j = 1, \ldots, n$
18:          Update moments accountant
19:          $q \leftarrow \frac{2 + \lambda|n_0 - n_1|}{4 \exp(\lambda|n_0 - n_1|)}$
20:          **for** $l = 1, \ldots, L$ **do**
21:             $\alpha(l) \leftarrow \alpha(l) + \min\{2\lambda^2 l(l+1), \log((1-q)\left(\frac{1-q}{1-e^{2\lambda}q}\right)^l + qe^{2\lambda l})\}$
22:          **end for**
23:          Update the student, $S$, using SGD
24:          $\nabla_{\theta_S} - \sum_{j=1}^{n} r_j \log S(\hat{\mathbf{u}}_j) + (1 - r_j) \log(1 - S(\hat{\mathbf{u}}_j))$
25:       **end for**
26:       Sample $\mathbf{z}_1, \ldots, \mathbf{z}_n \overset{\text{i.i.d.}}{\sim} P_{\mathcal{Z}}$
27:       Update the generator, $G$, using SGD
28:       $\nabla_{\theta_G} \left[ \sum_{i=1}^{n} \log(1 - S(G(\mathbf{z}_j))) \right]$
29:       $\hat{\epsilon} \leftarrow \min_l \frac{\alpha(l) + \log(\frac{1}{\delta})}{l}$
30:    **end for**
31: **end while**
32: **Output:** $G$

---

---

**Algorithm 4** Pseudo-code of PATE-GAN; `synthcity`

---

1: **Input:** $\delta, \mathcal{D}, n_T, n_S$, batch size $n$, number of teachers $k$, noise size $\lambda$
2: **Initialize:** $\theta_G, \theta_T^1, \ldots, \theta_T^k, \theta_S, \alpha(l) = 0$ for $l = 1, \ldots, L$
3: **Partition dataset into $k$ subsets $\mathcal{D}_1, \ldots, \mathcal{D}_k$ of size $\frac{|\mathcal{D}|}{k}$**
4: **while** $\hat{\epsilon} < \epsilon$ **do**
5:     **for** $t_2 = 1, \ldots, n_T$ **do**
6:         Sample $\mathbf{z}_1, \ldots, \mathbf{z}_n \overset{\text{i.i.d.}}{\sim} P_{\mathcal{Z}}$
7:         **for** $i = 1, \ldots, k$ **do**
8:             **Sample $\mathbf{u}_1, \ldots, \mathbf{u}_n \overset{\text{i.i.d.}}{\sim} \mathcal{D}_i$**
9:             **Update teacher, $T_i$, using SGD**
10:             $\nabla_{\theta_T^i} - \left[ \sum_{j=1}^d \log(T_i(\mathbf{u}_j)) + \log(1 - T_i(G(\mathbf{z}_j))) \right]$
11:         **end for**
12:     **end for**
13:     **for** $t_3 = 1, \ldots, n_S$ **do**
14:         Sample $\mathbf{z}_1, \ldots, \mathbf{z}_n \overset{\text{i.i.d.}}{\sim} P_{\mathcal{Z}}$
15:         **for** $j = 1, \ldots, n$ **do**
16:             $\hat{\mathbf{u}}_j \leftarrow G(\mathbf{z}_j)$
17:             $r_j \leftarrow \text{PATE}_\lambda(\hat{\mathbf{u}}_i)$ for $j = 1, \ldots, n$
18:             Update moments accountant
19:             $q \leftarrow \frac{2 + \lambda |n_0 - n_1|}{4 \exp(\lambda |n_0 - n_1|)}$
20:             **for** $l = 1, \ldots, L$ **do**
21:                 $\alpha(l) \leftarrow \alpha(l) + \min\{2\lambda^2 l(l+1), \log((1-q)\left(\frac{1-q}{1-e^{2\lambda}q}\right)^l + qe^{2\lambda l})\}$
22:             **end for**
23:             Update the student, $S$, using SGD
24:             $\nabla_{\theta_S} - \sum_{j=1}^n r_j \log S(\hat{\mathbf{u}}_j) + (1 - r_j)\log(1 - S(\hat{\mathbf{u}}_j))$
25:         **end for**
26:         Sample $\mathbf{z}_1, \ldots, \mathbf{z}_n \overset{\text{i.i.d.}}{\sim} P_{\mathcal{Z}}$
27:         Update the generator, $G$, using SGD
28:         $\nabla_{\theta_G} \left[ \sum_{i=1}^n \log(1 - S(G(\mathbf{z}_j))) \right]$
29:         $\hat{\epsilon} \leftarrow \min_l \frac{\alpha(l) + \log(\frac{1}{\delta})}{l}$
30:     **end for**
31: **end while**
32: **Output:** $G$

---

---

**Algorithm 5** Pseudo-code of PATE-GAN; turing

---

1: **Input:** $\delta, \mathcal{D}, n_T, n_S$, batch size $n$, number of teachers $k$, noise size $\lambda$
2: **Initialize:** $\theta_G, \theta_T^1, \ldots, \theta_T^k, \theta_S, \alpha(l) = 0$ **for** $l = 1, \ldots, L$
3: **Partition dataset into $k$ subsets $\mathcal{D}_1, \ldots, \mathcal{D}_k$ of size $\frac{|\mathcal{D}|}{k}$**
4: **while** $\hat{\epsilon} < \epsilon$ **do**
5:     **for** $t_2 = 1, \ldots, n_T$ **do**
6:         Sample $\mathbf{z}_1, \ldots, \mathbf{z}_n \overset{\text{i.i.d.}}{\sim} P_{\mathcal{Z}}$
7:         **for** $i = 1, \ldots, k$ **do**
8:             **Sample $\mathbf{u}_1, \ldots, \mathbf{u}_n \overset{\text{i.i.d.}}{\sim} \mathcal{D}_i$**
9:             **Update teacher, $T_i$, using SGD**
10:             $\nabla_{\theta_T^i} - \left[ \sum_{j=1}^d \log(T_i(\mathbf{u}_j)) + \log(1 - T_i(G(\mathbf{z}_j))) \right]$
11:         **end for**
12:     **end for**
13:     **for** $t_3 = 1, \ldots, n_S$ **do**
14:         Sample $\mathbf{z}_1, \ldots, \mathbf{z}_n \overset{\text{i.i.d.}}{\sim} P_{\mathcal{Z}}$
15:         **for** $j = 1, \ldots, n$ **do**
16:             $\hat{\mathbf{u}}_j \leftarrow G(\mathbf{z}_j)$
17:             $r_j \leftarrow \mathbf{PATE}_\lambda(\hat{\mathbf{u}}_i)$ **for** $j = 1, \ldots, n$
18:             **Update moments accountant**
19:             $q \leftarrow \frac{2 + \lambda|n_0 - n_1|}{4 \exp(\lambda|n_0 - n_1|)}$
20:             **for** $l = 1, \ldots, L$ **do**
21:                 $\alpha(l) \leftarrow \alpha(l) + \min\{2\lambda^2 l(l+1), \log((1-q)\left(\frac{1-q}{1-e^{2\lambda}q}\right)^l + qe^{2\lambda l})\}$
22:             **end for**
23:             Update the student, $S$, using SGD
24:             $\nabla_{\theta_S} - \sum_{j=1}^n r_j \log S(\hat{\mathbf{u}}_j) + (1 - r_j) \log(1 - S(\hat{\mathbf{u}}_j))$
25:         **end for**
26:         Sample $\mathbf{z}_1, \ldots, \mathbf{z}_n \overset{\text{i.i.d.}}{\sim} P_{\mathcal{Z}}$
27:         Update the generator, $G$, using SGD
28:         $\nabla_{\theta_G} \left[ \sum_{i=1}^n \log(1 - S(G(\mathbf{z}_j))) \right]$
29:         $\hat{\epsilon} \leftarrow \min_l \frac{\alpha(l) + \log(\frac{1}{\delta})}{l}$
30:     **end for**
31: **end while**
32: **Output:** $G$

---

---

**Algorithm 6** Pseudo-code of PATE-GAN; `borealis`

---

1: **Input:** $\delta, \mathcal{D}, n_T, n_S$, batch size $n$, number of teachers $k$, noise size $\lambda$
2: **Initialize:** $\theta_G, \theta_T^1, \ldots, \theta_T^k, \theta_S, \alpha(l) = 0$ for $l = 1, \ldots, L$
3: Partition dataset into $k$ subsets $\mathcal{D}_1, \ldots, \mathcal{D}_k$ of size $\frac{|\mathcal{D}|}{k}$
4: **while** $\hat{\epsilon} < \epsilon$ **do**
5:     **for** $t_2 = 1, \ldots, n_T$ **do**
6:         Sample $\mathbf{z}_1, \ldots, \mathbf{z}_n \overset{\text{i.i.d.}}{\sim} P_{\mathcal{Z}}$
7:         **for** $i = 1, \ldots, k$ **do**
8:             Sample $\mathbf{u}_1, \ldots, \mathbf{u}_n \overset{\text{i.i.d.}}{\sim} \mathcal{D}_i$
9:             Update teacher, $T_i$, using SGD
10:             $\nabla_{\theta_T^i} - \left[ \sum_{j=1}^{d} \log(T_i(\mathbf{u}_j)) + \log(1 - T_i(G(\mathbf{z}_j))) \right]$
11:         **end for**
12:     **end for**
13:     **for** $t_3 = 1, \ldots, n_S$ **do**
14:         Sample $\mathbf{z}_1, \ldots, \mathbf{z}_n \overset{\text{i.i.d.}}{\sim} P_{\mathcal{Z}}$
15:         **for** $j = 1, \ldots, n$ **do**
16:             $\hat{\mathbf{u}}_j \leftarrow G(\mathbf{z}_j)$
17:             $r_j \leftarrow \text{PATE}_\lambda(\hat{\mathbf{u}}_i)$ for $j = 1, \ldots, n$
18:             Update moments accountant
19:             $q \leftarrow \frac{2 + \lambda|n_0 - n_1|}{4 \exp(\lambda|n_0 - n_1|)}$
20:             **for** $l = 1, \ldots, L$ **do**
21:                 $\alpha(l) \leftarrow \alpha(l) + \min\{2\lambda^2 l(l+1), \log((1-q)\left(\frac{1-q}{1-e^{2\lambda}q}\right)^l + qe^{2\lambda l})\}$
22:             **end for**
23:             Update the student, $S$, using SGD
24:             $\nabla_{\theta_S} - \sum_{j=1}^{n} r_j \log S(\hat{\mathbf{u}}_j) + (1 - r_j) \log(1 - S(\hat{\mathbf{u}}_j))$
25:         **end for**
26:         Sample $\mathbf{z}_1, \ldots, \mathbf{z}_n \overset{\text{i.i.d.}}{\sim} P_{\mathcal{Z}}$
27:         Update the generator, $G$, using SGD
28:         $\nabla_{\theta_G} \left[ \sum_{i=1}^{n} \log(1 - S(G(\mathbf{z}_j))) \right]$
29:         $\hat{\epsilon} \leftarrow \min_l \frac{\alpha(l) + \log(\frac{1}{\delta})}{l}$
30:     **end for**
31: **end while**
32: **Output:** $G$

---

---

**Algorithm 7** Pseudo-code of PATE-GAN; `smartnoise`

---

1: **Input:** $\delta, \mathcal{D}, n_T, n_S$, batch size $n$, number of teachers $k$, noise size $\lambda$
2: **Initialize:** $\theta_G, \theta_T^1, \ldots, \theta_T^k, \theta_S, \alpha(l) = 0$ for $l = 1, \ldots, L$
3: Partition dataset into $k$ subsets $\mathcal{D}_1, \ldots, \mathcal{D}_k$ of size $\frac{|\mathcal{D}|}{k}$
4: **while** $\hat{\epsilon} < \epsilon$ **do**
5:     **for** $t_2 = 1, \ldots, n_T$ **do**
6:         Sample $\mathbf{z}_1, \ldots, \mathbf{z}_n \overset{\text{i.i.d.}}{\sim} P_{\mathcal{Z}}$
7:         **for** $i = 1, \ldots, k$ **do**
8:             Sample $\mathbf{u}_1, \ldots, \mathbf{u}_n \overset{\text{i.i.d.}}{\sim} \mathcal{D}_i$
9:             Update teacher, $T_i$, using SGD
10:             $\nabla_{\theta_T^i} - \left[ \sum_{j=1}^d \log(T_i(\mathbf{u}_j)) + \log(1 - T_i(G(\mathbf{z}_j))) \right]$
11:         **end for**
12:     **end for**
13:     **for** $t_3 = 1, \ldots, n_S$ **do**
14:         Sample $\mathbf{z}_1, \ldots, \mathbf{z}_n \overset{\text{i.i.d.}}{\sim} P_{\mathcal{Z}}$
15:         **for** $j = 1, \ldots, n$ **do**
16:             $\hat{\mathbf{u}}_j \leftarrow G(\mathbf{z}_j)$
17:             $r_j \leftarrow \text{PATE}_\lambda(\hat{\mathbf{u}}_i)$ for $j = 1, \ldots, n$
18:             Update moments accountant
19:             $q \leftarrow \frac{2 + \lambda |n_0 - n_1|}{4 \exp(\lambda |n_0 - n_1|)}$
20:             **for** $l = 1, \ldots, L$ **do**
21:                 $\alpha(l) \leftarrow \alpha(l) + \min\{2\lambda^2 l(l+1), \log((1-q)\left(\frac{1-q}{1-e^{2\lambda}q}\right)^l + qe^{2\lambda l})\}$
22:             **end for**
23:             Update the student, $S$, using SGD
24:             $\nabla_{\theta_S} - \sum_{j=1}^n r_j \log S(\hat{\mathbf{u}}_j) + (1 - r_j) \log(1 - S(\hat{\mathbf{u}}_j))$
25:         **end for**
26:     Sample $\mathbf{z}_1, \ldots, \mathbf{z}_n \overset{\text{i.i.d.}}{\sim} P_{\mathcal{Z}}$
27:     Update the generator, $G$, using SGD
28:     $\nabla_{\theta_G} \left[ \sum_{i=1}^n \log(1 - S(G(\mathbf{z}_j))) \right]$
29:     $\hat{\epsilon} \leftarrow \min_l \frac{\alpha(l) + \log(\frac{1}{\delta})}{l}$
30:     **end for**
31: **end while**
32: **Output:** $G$

---

