# OpenReview forum: "The Elusive Pursuit of Reproducing PATE-GAN: Benchmarking, Auditing, Debugging"
_TMLR — Accepted by TMLR_

### Review · Reviewer_Geuw · 2024-10-24

**Summary Of Contributions:**

This work empirically evaluates a number of publicly available implementations of PATE-GAN implementations. The work evaluates both the performance claims made in the respective works as well as an empirical privacy audit of the frameworks.

**Audience:**

Yes

**Broader Impact Concerns:**

Already covered in the work.

**Claims And Evidence:**

Yes

**Requested Changes:**

While I understand that these were the datasets described in the works making for a fair comparison, I would like to see how well (or how badly) the privacy-utility and the empirical epsilon estimation behave in settings which are a) more balanced with respect to the labels and b) use more complex underlying architectures and c) an additional epsilon.

One of the recent auditing frameworks introduced by Zanella-Beguelin, 2023 (cited in this work) significantly improves on the reported privacy bounds compared to the Clopper-Pearson approach, so it would be advisable to use the method proposed in this work for epsilon bound estimation.

**Strengths And Weaknesses:**

This is a very timely paper showing the problems with existing DP approaches to PATE-GAN. The scale of the issue becomes even more striking, given that these frameworks were proposed by many different author groups, all suffering from different issues, but ultimately passing the associated risks to the end users whose data can be misused at the end. Therefore I can safely say that this work is of importance and of interest to the community (and not just the privacy community).

I found no fundamental issues in this work, but have made a few additional requests in the comments below in order to improve the privacy bound estimation and show how these models perform in more realistic deep learning scenarios. Given that MIA results are already bad enough, I find the choice of attacks to be decent too and do not have any additional requests here. In general I find the work to be sound in the choice of techniques and methods.

One potential weakness for me is that I am still uncertain as to why is the loss of performance so drastic compared to the ones reported in the original papers? Is this purely because of the bugs or am I missing something? Because I find the 50+% to be extremely worrying and would like to clarify if this performance difference is to do with software issues, rather than misreported results in the original papers which were evaluated.

---

> ### Author Response · Authors · 2024-11-26
> **Authors Response to Reviewer `Geuw` (1/2)**
>
> We thank the reviewer for their helpful comments and suggestions. In the Overall Message, we summarize the changes in the updated submission and common concerns. Below, we respond to specific questions/concerns:
>
>
> > One potential weakness for me is that I am still uncertain as to why is the loss of performance so drastic compared to the ones reported in the original papers? Is this purely because of the bugs or am I missing something? Because I find the 50+% to be extremely worrying and would like to clarify if this performance difference is to do with software issues, rather than misreported results in the original papers which were evaluated.
>
> As explained in the Overall Message, the main contributing factor for the utility drop of `original` compared to the results in the original paper is that it expects the distribution of the data labels to be provided at generation (which constitutes unaccounted privacy leakage). In the updated submission, we confirmed this hypothesis by running an additional experiment (see Table 6), effectively reproducing the results reported in the original paper (average utility drop changes from -24.57% to -3.34%).
>
>
> > I would like to see how well (or how badly) the privacy-utility and the empirical epsilon estimation behave in settings which are a) more balanced with respect to the labels and b) use more complex underlying architectures and c) an additional epsilon.
>
> In the updated submission, we selected MNIST because it is: 1) more balanced – all labels are represented approximately equally, and 2) is more complex compared to tabular datasets as it requires modeling high levels of correlations. We ran additional utility benchmarking with this dataset and all six implementations over numerous epsilons and reported the results in Appendix B. Unfortunately, this experiment is still running as it is computationally intensive, but we are confident we can include it before the end of the discussion period.
>
> Unfortunately, it is computationally infeasible to execute the auditing experiments for MNIST as, for some implementations, it takes numerous hours to fit a single model, while we require training 2,000 models (1,000 w/ the target and 1,000 w/out) for each implementation.
>
>
> >  One of the recent auditing frameworks introduced by Zanella-Beguelin, 2023 (cited in this work) significantly improves on the reported privacy bounds compared to the Clopper-Pearson approach, so it would be advisable to use the method proposed in this work for epsilon bound estimation.
>
> In the updated submission, we ran additional experiments for both the worst-case dataset and Kaggle Cervical Cancer and reported the results in Section 6.2 (also see Figure 11 in Appendix B). Similarly to the Clopper-Pearson method, we always get higher empirical estimates than 1 and even achieve slightly higher estimates on 9 out of 12 occasions.

---

> > ### Author Response · Authors · 2024-12-03
> > **Authors Response to Reviewer `Geuw` (2/2)**
> >
> > We have updated the submission with an additional change:
> >
> > > I would like to see how well (or how badly) the privacy-utility and the empirical epsilon estimation behave in settings which are a) more balanced with respect to the labels and b) use more complex underlying architectures and c) an additional epsilon.
> >
> > In the updated submission, we ran additional utility benchmarking on MNIST and reported the results in Appendix B. Overall, none of the implementations demonstrate promising results on this dataset. While `original` and `turing` outperform the others, as previously discussed, neither correctly implements PATE. It is important to note that none of the implementations are specifically designed for image datasets, leaving significant room for potential improvements.

---

### Review · Reviewer_jP3A · 2024-11-10

**Summary Of Contributions:**

The paper evaluates the reproducibility, privacy, and utility claims of PATE-GAN, by benchmarking six available PATE-GAN implementations against the resutls reported at the original paper. The contributions are as follows:

1) The paper benchmarks six PATE-GAN implementations to reproduce utility metrics such as AUROC scores, demonstrating that none reproduce the original model's performance and often underperform.

2) The paper presents an in-depth privacy audit using Differential Privacy (DP) techniques and discover significant privacy leaks in all six implementations.

3) The paper identifies a series of privacy and implementation bugs, including 17 privacy violations and various issues affecting the DP guarantees.

4) The code is open sourced to the research community.

**Audience:**

Yes

**Claims And Evidence:**

Yes

**Requested Changes:**

1) Could you please report the experimental results for figures 1 and 3 over multiple runs? what is the mean and standard deviation?

2) It might be beneficial to add a section that addresses the ethical implications of using the DP models with the privacy violation in real-world applications, especially in applications such as healthcare or finance.

3) Table 1 reports the reduction in AUROC from the original paper (Jordon et al., 2019). The drop is significant (from 42% to 80%). Are these results by running their code (even with the privacy bugs)? How can you explain this drop?

4) Please mention what is the orange dotted line in Figure 3. Is it the same as the orange solid line?

5) How the number of MIA and the number of collection datapoints was selected (at page 10 of the paper) ?

6) Would it be possible to perform the DP auditing with different black-box MIAs (Figure 5) on the rest of the datasets (apart from the Kaggle Cervical Cancer ) ? What happens in the case when the theoretical epsilon is greater than 1? Do we observe privacy violation? by what margin?

**Strengths And Weaknesses:**

Strengths:

1) The paper presents a detailed study of PATE-GAN implementations with respect to utility and privacy.

2) The use of privacy auditing techniques, including membership inference attacks (MIA), demonstrates substantial privacy risks in these implementations.

3) The paper  identifies the privacy violations and provides insights into the bugs and implementation errors that result to them.

4) The open soursing of the code facilitates future research.

Weaknesses:

Some parts of the paper are not explained properly and are not clear. Please see the Requested Changes section for more details.

---

> ### Author Response · Authors · 2024-11-26
> **Authors Response to Reviewer `jP3A`**
>
> We thank the reviewer for their helpful comments and suggestions. In the Overall Message, we summarize the changes in the updated submission and common concerns. Below, we respond to specific questions/concerns:
>
>
> > 1. Could you please report the experimental results for figures 1 and 3 over multiple runs? what is the mean and standard deviation?
>
> Figure 1 does include scores over multiple runs – similarly to the original paper, we report the best AUROC scores over 25 synthetic datasets (we train five models and generate five synthetic datasets per model). In Figure 10, we replot the average scores and the standard errors. In the updated submission, we also reported the mean (instead of maximum score) and standard error of UCI Epileptic Seizure in Figure 8, but, as expected, we did not reach the scores reported in the original paper even within the standard deviation.
>
> In the updated submission, we also reported the experiment shown in Figure 3 over ten training runs and plotted the average and standard error in Figure 10, which yields the same trends.
>
>
> > 2. It might be beneficial to add a section that addresses the ethical implications of using the DP models with the privacy violation in real-world applications, especially in applications such as healthcare or finance.
>
> In the updated submission, we added a new paragraph in Section 7 highlighting the importance of deploying correct DP models.
>
>
> > 3. Table 1 reports the reduction in AUROC from the original paper (Jordon et al., 2019). The drop is significant (from 42% to 80%). Are these results by running their code (even with the privacy bugs)? How can you explain this drop?
>
> We are using the implementations without modifying them (without fixing any privacy bugs), as they are adopted and used in practice. As explained in the Overall Message, the main contributing factor for the utility drop of `original` compared to the results in the original paper is that it expects the distribution of the data labels to be provided at generation (which constitutes unaccounted privacy leakage). In the updated submission, we confirmed this hypothesis by running an additional experiment (see Table 6), effectively reproducing the results reported in the original paper (average utility drop changes from -24.57% to -3.34%).
>
>
> > 4. Please mention what is the orange dotted line in Figure 3. Is it the same as the orange solid line?
>
> The orange dotted and solid lines are the same line. Note that `synthcity` only needs 17 iterations to converge, while the other implementations take much longer; hence, the dotted lines appear solid.
>
>
> > 5. How the number of MIA and the number of collection datapoints was selected (at page 10 of the paper) ?
>
> As discussed in Section 6.2, we use two state-of-the-art MIAs proposed for tabular data – GroundHog and Querybased. We use Querybased on a worst-case dataset, which we manually construct (including the target record) and GroundHog on Kaggle Cervical Cancer. For Kaggle Cervical Cancer, we run a mini-MIA to select the most vulnerable record and then run the whole attack on all six implementations. For both, we train 1,000 models on the “in” dataset and 1,000 on the “out” dataset (i.e., 2,000 shadow models in total); we use 40% for fitting the Random Forest classifier, 20% for validation, and 40% for testing and empirically estimating epsilon.
>
>
> > 6. Would it be possible to perform the DP auditing with different black-box MIAs (Figure 5) on the rest of the datasets (apart from the Kaggle Cervical Cancer) ? What happens in the case when the theoretical epsilon is greater than 1? Do we observe privacy violation? by what margin?
>
> Since DP guarantees apply in all cases, i.e., they are meant to hold in the worst case, we have intentionally selected small and unfavorable scenarios, following previous work on DP auditing (Nasr et al., 2021; 2023; Lokna et al., 2023; Annamalai et al., 2024b). This allows us to efficiently train 2,000 models (1,000 w/ the target and 1,000 w/out) for each implementation. Unfortunately, it is computationally infeasible to do so for the other datasets.
>
> Since our auditing experiments demonstrate privacy violations with epsilon=1 (and we further confirm this by manually inspecting the codebases), we respectfully believe additional experiments would not add much to the paper as we are confident that privacy violations for other epsilon values will also be detected. If one had the computational resources to train more models (say 20,000), the margin between estimated and theoretical epsilon would also increase, as demonstrated in previous studies (Annamalai et al., 2024b).

---

### Review · Reviewer_ztgz · 2024-11-21

**Summary Of Contributions:**

This work provides a benchmark evaluation of six popular implementations of PATE-GAN [1]. The authors showed that the all six implementation leaks more privacy than intended ($\epsilon_{emp} >> \epsilon $). The goal of teh study is to reproduce the utility performance reported in the original paper, and using membership inference as auditor to estimate PATE-GAN’s privacy guarantees. Among other findings, they identified 17 privacy violations and 5 other bugs based on how PATE is implemented.


[1] James Jordon, Jinsung Yoon, and Mihaela Van Der Schaar. PATE-GAN: Generating synthetic data with differential privacy guarantees. In ICLR, 2019.

**Audience:**

Yes

**Claims And Evidence:**

Yes

**Requested Changes:**

My main worry is that I see the author's making exactly the same mistakes that the previous implementations made. Specifically, I would love to see the following analysis to justify the claim of this paper.
1. Can the authors use exactly the same datasets, framework version e.g Tensoflow / Pytorch and python version, and report the results?
2. While I am in favor of this paper provided 1 above is verified, rather than releasing the code, I would like to see a docker file. Such that the experiments can be reproduced.
3. The claims of the paper are really concerning and interesting. For instance, "We find that original and turing do not actually implement the PATE framework or a moments accountant".  As also pointed out by the authors, "This is a serious deviation that can compromise privacy". I am not surprised the bugs are not fixed after contacting the authors, but I am interested in the response of the authors after you pointed out the errors.
4. I don't understand the rationale behind using different hyperparameters for the same work.  "Also, all implementations have different network depths and noise dimensions, mostly depending on the input data." Why would different authors have different implementations of the same work, especially in terms of the hyperparameters? I find this strange.

5. One fundamental problem between the original paper [1] and your reimplementations is that there is no standard deviation reported. Could it be that the identified discrepancies is within the standard deviation? For instance, I find the results of original and borealis not to be entirely far apart.

6. Rather than the claim that there are discrepancies in the results. As pointed out by the authors in "Data Support and Processing", the data processing differs across different implementations. So shouldn't the data processing be one / most important reason for the discrepancies? I will assume that each of the other implementations adapts the original implementation for fair comparison. If this is the case, I will expect some discrepancies as adaptation of an original code could lead to different result.

7. Can the authors provide more details on the MI attacks (Querybased and GroundHog). How was it used. How many shadow models were used? etc. This saves the time in reading each of the papers.


[1] James Jordon, Jinsung Yoon, and Mihaela Van Der Schaar. PATE-GAN: Generating synthetic data with differential privacy guarantees. In ICLR, 2019.

**Strengths And Weaknesses:**

**Strengths:**
- This work discover numerous privacy violations across all six implementations of PATE-GAN
- The authors did a good job in auditing the implementations using MI attacks
- They showed that the empirical epsilon is greater than the theoretical epsilon


***Weaknesses:***
There is one major weakness with this study.

While I am not entirely convinced, (see requested changes section), the basic experiments of reproducing the results of the paper as is, is missing. Specifically, what would have been the results when the code of the six implementations are run according to the original paper, including the hyperparameters?

To buttress that, since several implementation use different hyperparameters as shown in Table 3, surely, the results will differ. Hence, the need for a uniform assessment across all implementations using the same hyperparameter.

---

> ### Author Response · Authors · 2024-11-26
> **Authors Response to Reviewer `ztgz ` (1/2)**
>
> We thank the reviewer for their helpful comments and suggestions. In the Overall Message, we summarize the changes in the updated submission and common concerns. Below, we respond to specific questions/concerns:
>
> > Specifically, what would have been the results when the code of the six implementations are run according to the original paper, including the hyperparameters?
> > To buttress that, since several implementation use different hyperparameters as shown in Table 3, surely, the results will differ. Hence, the need for a uniform assessment across all implementations using the same hyperparameter.
>
> As stated in Section 4, one of our main goals is to benchmark the six implementations as they are released, without modifying them, since this is how they are adopted and used in practice. Furthermore, we can assume that the authors followed the common practice of releasing the implementations with the best choice of architecture and set of hyperparameters after conducting their own hyperparameter search. Testing the default hyperparameters is also common in DP benchmark studies (Tao et al., 2022; McKenna et al., 2022; Stadler et al., 2022, Ganev et al., 2024).
>
> Nevertheless, in the updated submission, we ran additional utility experiments for all implementations using the same hyperparameters and architectures as the ones specified in the original paper, with epsilon=1 on all four datasets (please see Section 5 as well as Table 12 in Appendix B). Our hypothesis is proven correct, as the the utility of all implementations drops by an average of 17.5% compared to when their respective default hyperparameters are used.
>
>
> > 1. Can the authors use exactly the same datasets, framework version e.g Tensoflow / Pytorch and python version, and report the results?
> > 2. While I am in favor of this paper provided 1 above is verified, rather than releasing the code, I would like to see a docker file. Such that the experiments can be reproduced.
>
> We did not change any hyperparameters (or other settings, dependencies, etc.) that could reduce the performance of the implementations. We also use exactly the same datasets from the original paper. Respectfully, focusing on details like the exact python/framework version might overlook the main goal of our paper, i.e., testing the utility/privacy of the six implementations in the way they are widely used in practice. Incidentally, there is no reason to believe that using incrementally different versions of Python and frameworks such as Tensoflow and Pytorch would lead to _significantly_ different results as these open-source tools have millions of users/contributors and any significant discrepancy would break numerous live systems and would quickly be detected and fixed.
>
> Moreover, `original` does not even have an accompanying requirements file, which makes it impossible to infer the exact versions at release. Two of the implementations (`synthcity` and `smartnoise`) are active python packages, which are continuously being updated, including the versions of their requirements. Also, when setting up the environment for our experiments, there was a single minor conflict between `synthcity` and `smartnoise` (e.g., they require different versions of the package `opacus` but neither of them actually use it, as they manually implement the PATE-GAN specific moments accountant); in other words, we use explicitly allowed dependencies across all implementations.
>
> We would also like to highlight that even if there were a way to extract the exact Python/frameworks versions used at release time, this would not fix the privacy-related and other bugs in these implementations, and truly reflect their utility performance. For instance, `original` and `turing` do not even implement PATE, `original` requires the labels distribution at generation, `updated` and `borileas` do not generate synthetic data in the scale of the original data, etc.
>
>
> > 3. I am not surprised the bugs are not fixed after contacting the authors, but I am interested in the response of the authors after you pointed out the errors.
>
> So far, the authors have not responded to our emails/GitHub issues, with a single exception (the authors of `turing` said they would be happy to adopt a more modern implementation of PATE-GAN). As mentioned in Section 7, we will include the GitHub issues we raised in the camera-ready version (we did not include the links to preserve submission anonymity).

---

> ### Author Response · Authors · 2024-11-26
> **Authors Response to Reviewer `ztgz ` (2/2)**
>
> > 4. Why would different authors have different implementations of the same work, especially in terms of the hyperparameters? I find this strange.
>
> This is indeed a key factor driving the motivation behind our work. On the one hand, different authors likely optimize their implementations (choice of architecture, hyperparameters, etc.) over different datasets, thus choosing different optimal hyperparameters. On the other hand, we find it intriguing that the same authors have released three versions of the same model, all with different architectures and hyperparameters, as shown in Table 3.
>
>
> > 5. One fundamental problem between the original paper [1] and your reimplementations is that there is no standard deviation reported. Could it be that the identified discrepancies is within the standard deviation? For instance, I find the results of original and borealis not to be entirely far apart.
>
> Please note that the original paper does not report standard deviations but a single score.
>
> As discussed in Section 5, we follow `updated` (released by the original authors) and report the best score over 25 synthetic datasets (we train five models and generate five synthetic datasets per model). In the updated submission, we also reported the mean (instead of maximum score) and standard error of UCI Epileptic Seizure in Figure 8, but, as expected, we did not reach the scores reported in the original paper even within the standard deviation. Furthermore, in the updated submission, we ran another experiment in which we provided the labels distribution at generation to `original` (as the original authors do) and effectively managed to reproduce the results reported in the original paper (average utility drop changes from -24.57% to -3.34%).
>
> Finally, while in Table 4, `borealis`’s performance might look not too far from the one reported in the original paper, Figure 1 shows that its utility severely degrades for epsilons larger than 1.
>
>
> > 6. As pointed out by the authors in "Data Support and Processing", the data processing differs across different implementations. I will assume that each of the other implementations adapts the original implementation for fair comparison. If this is the case, I will expect some discrepancies as adaptation of an original code could lead to different result.
>
> As stated in Section 5, we assume that the architecture, hyperparameters, and data processing are part of the model itself and that the authors have made the optimal decisions for their corresponding implementation before releasing it. To our knowledge, none of the authors of the newly proposed implementations actually compare their performance with `original`, another factor driving our work’s motivation.
>
>
> > Can the authors provide more details on the MI attacks (Querybased and GroundHog). How was it used. How many shadow models were used?
>
> Please note that while we discuss how DP auditing via MIAs works in general in Section 2, in Section 6.2 (Adversarial Model and Setup & Hyperparameters), we explain how the specific MIAs (Querybased and GroundHog) work in detail, i.e., how they extract features from the synthetic datasets. For both, we train 1,000 models on the “in” dataset and 1,000 on the “out” dataset (i.e., 2,000 shadow models in total). We use 40% for fitting the Random Forest classifier, 20% for validation, and 40% for testing and empirically estimating epsilon.

---

### Author Response · Authors · 2024-11-26
**Overall Message -- Summary of Changes and Common Comments**

We thank the reviewers for their helpful comments and suggestions.

## Summary of Changes
We have updated our submission with the requested changes, denoted in red in the new PDF.  Due to time constraints, we are submitting our rebuttal and the revised paper while still waiting for one additional experiment to complete (it takes a couple of weeks to run).

The main changes in this revision include:
* Section 1 – Updated Table 1 to denote the utility drop from the reported utility in the original paper rather than from the utility on the real data (`ztgz`, `jP3A`, `Geuw`).
* Section 1 – Updated Table 2 to include an additional privacy violation, referred to as “labels distribution,”  where the exact label distribution of the data is leaked, which explains the utility drop in our evaluation of `original` (`ztgz`, `jP3A`, `Geuw`).
* Section 5 – Replotted Figure 1 (see Figure 8 in Appendix B) to show the mean and standard error of AUROC instead of the maximum score (`ztgz`, `jP3A`).
* Section 5 – Ran additional utility experiments using the `original` implementation using the distribution of labels in the dataset (which constitutes unaccounted privacy leakage) with epsilon=1 on all four datasets to more accurately reproduce the utility performance in the original paper. The results are presented in Table 6; please also see the Comment on the Utility Drop below (`ztgz`, `jP3A`, `Geuw`).
* Section 5 – Ran additional utility experiments using the same hyperparameters and architectures as specified in the original paper with epsilon=1 on all four datasets; Results are presented in Table 12 in Appendix B (`ztgz`).
* Section 6.1 – Replotted Figure 3 (see Figure 10 in Appendix B) to show the mean and standard error of the moments (`jP3A`).
* Section 6.2 – Reran the auditing experiments with  Zanella-Beguelin et al.’s method of empirically estimating epsilon; Results are presented in Figure 11 in Appendix B (`Geuw`).
* Section 7 – Added a paragraph discussing the importance of ensuring robust privacy protections in real-world applications of DP models (`jP3A`).

Change to be made once the last experiment is complete (we’re confident before the end of the discussion period):
* Appendix B – We are running an additional utility experiment on a more complex dataset with balanced labels – namely, MNIST – across a range of epsilons (`Geuw`).


## Comment on the Utility Drop for All Implementations
The only common comment shared by all three reviewers (`ztgz`, `jP3A`, `Geuw`) is the surprising utility drop all six implementations suffer compared to the reported performance in the original paper. In our updated submission, we provided further clarification in Section 5, more precisely:
1. We report _normalized_ utility drop, using the majority AUROC score of 0.5 as a baseline (i.e., if AUROC drops from  87% to 50%, we count this as a 100% drop). In our updated submission, we adjusted the utility drops in Table 1;
2. All four datasets are quite imbalanced, with the proportion of positive labels ranging from only 0.17% to 20% (see Table 7 in Appendix A). Applying DP noise causes a large disparate utility drop, which is captured by AUROC – for instance, most implementations have random or close to random performance on Kaggle Cancer (which has the highest imbalance of 0.17%);
3. One of the main reasons for the utility degradation seen in our paper is that the `original` implementation uses the exact distribution of labels in the training dataset when generating synthetic data, which constitutes unaccounted privacy leakage. This prompts us to fix this issue in our work. To confirm this hypothesis, we ran an additional experiment in our updated revision using the `original` implementation and the exact distribution of labels with epsilon=1 on all four datasets (see Table 6). The average utility drop changes from -24.57% to -3.34%, effectively reproducing the performance reported in the original paper.

---

> ### Author Response · Authors · 2024-12-03
> **Overall Message Follow-up -- Summary of Changes**
>
> ## Summary of Changes
>
> We have updated the submission with an additional change:
> * Appendix B – We ran an additional utility experiment on a more complex dataset with balanced labels – namely, MNIST – across a range of epsilons (`Geuw`).

---

### Decision · Action_Editor_ZV7Q · 2025-01-09

**Recommendation:** Accept with minor revision

**Comment:**

Many of the concerns by the reviewers have been addressed during the review process such running the experiments of the original paper (comment by reviewer ztgz) and considering more balanced datasets for experiments (comment by reviewer Geuw).

I think the paper does great job in revealing inconsistencies in the results of previous PATE-GAN papers and in finding bugs in the existing implementations. Although there were some concerns from the reviewer ztgz about how solid the experiments are, I think they are very valuable and as requested, the authors have run the experiments of the original paper as well to check the methods' utilities. Similarly to the reviewer Geuw, I am flagging the Reproducibility Certification. The code for the comparisons can be found in the supplements and I sincerely hope the authors will release a code repository as they mention in the abstract. I also hope you can address the following questions.

A question: What are exactly the privacy violations in 'smartnoise' and 'borealies' that you mention? If I understood correctly, there are those $q$-dependent terms missing in the moments accountant (Alg. 6 and 7 in your paper). But leaving those terms out would only give overestimates of the epsilons, and thus you'd get valid DP bounds if everything else is correct? I just don't understand where do the privacy violations mentioned in Table 2 come from.

Related questions and comments: Did you use versions of 'smartnoise' and 'borealies' where you modified the moments accountant? If not, then what do you think, how is it possible that the empirical epsilons are larger than the theoretical ones since there do not seem to be other obvious bugs? If you used the modified moments accountant, then did you check that all the conditions of the underlying privacy analysis are met? It seems to me that the original work by [Jordron et al.](https://openreview.net/pdf?id=S1zk9iRqF7) is overlooking, e.g., the condition on $q$ that is stated in the original PATE paper by [Papernot et al.](https://arxiv.org/pdf/1610.05755). Also, looking at the original PATE analysis, should the Laplace noise parameter $\lambda$ in the moments accountant of PATE-GAN be replaced by $\lambda^{-1}$? I am mentioning these since I think it would be nice to get understanding on why the empirical epsilons are so large, especially for 'smartnoise' and 'borealies'. In any case this is a nice illustration of the usefulness of privacy auditing.

I still would recommend the authors to go over the writing carefully which was also commented by the reviewer jP3A. In particular:

- I think it would be good to add details about the PATE method, like what is the meaning of the parameter $\lambda$, how is the Laplace noise scaled with it? (seems to be different scalings in PATE-GAN and PATE papers)

- I could not find the meaning of the variables $n_0$ and $n_1$ used in the pseudocodes, I had to look up from the original paper. Please add those and desribe all the other variables as well in the pseudocodes ($L$ etc.).

**Audience:**

Yes, the topic of the paper (privacy-preserving machine learning, synthetic data) fits very well to TMLR.

**Claims And Evidence:**

The paper studies six different implementations of the PATE-GAN, a popular differentially private GAN method for generating synthetic data with DP guarantees, which is based on the PATE (private aggregation of teacher ensembles, Papernot et al., 2018). The authors find several bugs in all of the six the implementations, cannot match the utility reported in previous works and moreover, by carrying out privacy auditing methods, find out that the empirical epsilon-values are bigger than the theoretical epsilon values obtained using the six implementations. For reproducibility, the authors are planning the make the implementation of these evaluations public.

---

> ### Author Response · Authors · 2025-02-10
> **Authors Response to Action Editor `ZV7Q `**
>
> We would like to thank the Action Editor and reviewers for their valuable feedback and suggestions, which helped us to significantly improve our paper. We also appreciate the recognition of our paper with the Reproducibility Certification.
>
> > What are exactly the privacy violations in `smartnoise` and `borealis` that you mention (in Table 2)?
>
> In Table 2, we list two types of privacy violations and one additional bug in `borealis`, as well as one privacy violation in `smartnoise` (all raised issues are listed in Appendix D).
> * `borealis`: i) a missing log operator in the second term of the moments accountant (see [issue 11](https://github.com/BorealisAI/private-data-generation/issues/11)), ii) direct metadata extraction from the input data, and iii) a missing post-processing step to scale the generated synthetic data back to the scale of the input data (see [issue 12](https://github.com/BorealisAI/private-data-generation/issues/12) for combined ii) and iii)).
> * `smartnoise`: the same missing operator in the moments accountant as i) in `borealis` (see [issue 596](https://github.com/opendp/smartnoise-sdk/issues/596)).
>
> From our empirical experiments on the Kaggle Cervical Cancer dataset, we observe that the missing log operator mostly affects the two implementations when they are trained for numerous iterations, causing the moments accountant to overestimate $\epsilon$ (see Figure 4).
>
> > Did you use versions of `smartnoise` and `borealis` where you modified the moments accountant?
>
> For all experiments, including auditing, we do not modify the implementations (unless specifically stated otherwise), nor do we correct any of the identified privacy violations or bugs. As noted in Section 5, to maintain consistency with other implementations and the original paper, we do not extract the bounds in a DP manner for `smartnoise`, saving its budget for model training.
>
>
> > If not, then what do you think, how is it possible that the empirical epsilons are larger than the theoretical ones since there do not seem to be other obvious bugs?
> > If you used the modified moments accountant, then did you check that all the conditions of the underlying privacy analysis are met?
>
> We provide our best intuition why $\epsilon_{emp} > \epsilon$ for `borealis` and `smartnoise`.
> * Worst-case dataset with a manually crafted record (Figure 5a): The excessive privacy leakage can likely be attributed to the implementations directly extracting metadata from the input data, as the target record falls outside the domain of the (0, 0, 0) records.
> * Average-case dataset with a real target (Figure 5b): While we cannot be entirely certain, we speculate that the excessive privacy leakage stems from differences in the privacy analysis between PATE-GAN and PATE (Papernot et al., 2017). In the camera-ready version, we have included an additional section (see Appendix A2) detailing five key discrepancies between the two. Compared to PATE, on which PATE-GAN is directly based, PATE-GAN:
>   * i) understates the privacy budget of a single query to the PATE mechanism by a factor of 2,
>   * ii) defines the noise parameter $\lambda$ initially as the inverse to the one in PATE (Papernot et al., 2017) but then follows the same privacy analysis as PATE,
>   * iii) omits a condition for q in the moments accountant,
>   * iv) takes a minimum over two factors instead of three in the moments accountant,
>   * v) releases the unperturbed $\epsilon$, whereas PATE (Papernot et al., 2017) releases a DP estimate of $\epsilon$.
>
> However, as noted in the paper, we leave the analysis and resolution of these discrepancies for future work.
>
>
> > I think it would be good to add details about the PATE method i) like what is the meaning of the parameter $\lambda$ (how is the Laplace noise scaled with it), ii) the meaning of the variables $n_0$ and $n_1$, iii) describe all the other variables.
>
> We added further details about all PATE parameters and provided insights into the balance between the noise parameter $\lambda$ and the number of teachers $k$  in Section 2.